# R-Horizon: How Far Can Your Large Reasoning Model Really Go in Breadth and Depth?

**Yi Lu**[1,2]    **Jianing Wang**[2*]    **Linsen Guo**[2]    **Wei He**[1,2]    **Hongyin Tang**[2]
**Tao Gui**[1,3*]    **Xuanjing Huang**[1,3]    **Xuezhi Cao**[2]    **Wei Wang**[2*]    **Xunliang Cai**[2]

[1]Fudan University    [2]Meituan LongCat Team
[3]Shanghai Collaborative Innovation Center of Intelligent Visual Computing

https://github.com/meituan-longcat/R-HORIZON

## Abstract

Recent trends in test-time scaling for reasoning models (e.g., OpenAI o1, DeepSeek-R1) have led to remarkable improvements through long Chain-of-Thought (CoT). However, existing benchmarks mainly focus on immediate, single-horizon tasks, failing to adequately evaluate models' ability to understand and respond to complex, long-horizon scenarios. To address this incomplete evaluation of Large Reasoning Models (LRMs), we propose R-Horizon, a method designed to stimulate long-horizon reasoning behaviors in LRMs through query composition. Based on R-Horizon, we construct a long-horizon reasoning Benchmark, comprising complex multi-step reasoning tasks with interdependent problems that span long reasoning horizons. Through comprehensive evaluation of LRMs using the R-Horizon Benchmark, we find that even the most advanced LRMs suffer significant performance degradation. Our analysis reveals that LRMs exhibit limited effective reasoning length and struggle to allocate thinking budget across multiple problems appropriately. Recognizing these limitations, we use R-Horizon to construct long-horizon reasoning data for reinforcement learning with verified rewards (RLVR). Compared to training with single-horizon data, RLVR with R-Horizon not only substantially improves performance on the multi-horizon reasoning tasks, but also promotes accuracy on standard reasoning tasks (+7.5 on AIME2024). These results position R-Horizon as a scalable, controllable, and low-cost paradigm for enhancing and evaluating the long-horizon reasoning capabilities of LRMs.

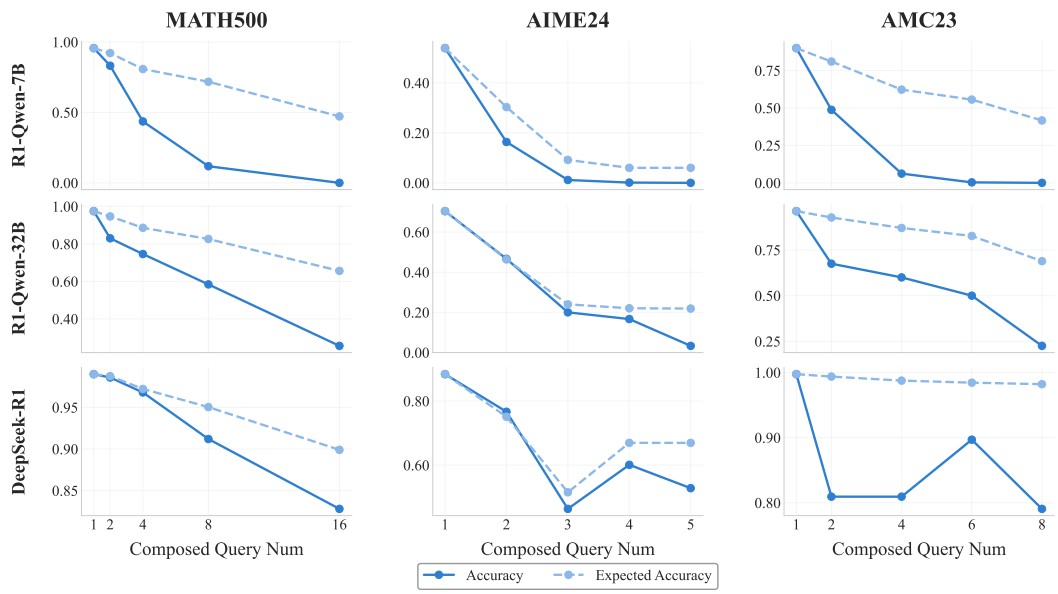

Figure 1: Actual versus expected accuracy of R1-series models on R-Horizon datasets.

*Corresponding authors. lygwjn@gmail.com, tgui@fudan.edu.cn, wangwei432@meituan.com.

## 1 INTRODUCTION

Recent advances in reasoning-focused language models, exemplified by OpenAI's o1 (OpenAI et al., 2024) and DeepSeek-R1 (Guo et al., 2025), have established test-time scaling as a fundamental component for enhancing reasoning abilities in large reasoning models (LRMs). Specifically, test-time scaling enables long Chain-of-Thought (CoT) (Wei et al., 2023) and induces sophisticated reasoning behaviors, leading to remarkable improvements on challenging reasoning tasks like mathematical reasoning (He et al., 2025b; Yu et al., 2025; Yue et al., 2025; Zeng et al., 2025), code generation (Luo et al., 2025a; Zeng et al., 2025) and agentic tasks (Team et al., 2025b;a).

By continuously expending computational resources throughout the reasoning process, models with longer reasoning trajectories achieve superior performance on various reasoning benchmarks (Muennighoff et al., 2025), yet this simultaneously exposes critical limitations in current training and evaluation paradigms. Existing training and evaluation datasets (Cobbe et al., 2021; Hendrycks et al., 2021; Jain et al., 2024) primarily confine themselves to the reasoning of isolated problems, focusing on immediate single-horizon tasks where questions and answers remain independent of each other. However, real-world scenarios often require an AI agent to reason, plan, and act over an extended series of steps, sometimes thousands or even millions, where inference must span across multiple sequential and potentially interdependent problems (Yao et al., 2024; Tao et al., 2025). Single-horizon evaluation paradigms cannot effectively assess the ability of a model to understand and respond to complex, multi-horizon tasks or scenarios that require a sequence of logical steps over a longer period of time. Moreover, conventional reinforcement learning (RL) typically focuses on single, isolated problems, preventing models from developing long-horizon reasoning capabilities to tackle multiple problems through the RL process. The incomplete picture of training and evaluation paradigms raises a fundamental question: *How far can large reasoning models really go in breadth and depth?*

In this study, we propose R-HORIZON, a simple yet effective method to stimulate long-horizon reasoning behaviors in LRMs through query composition. This method aims to construct dependencies and concatenate existing single-horizon tasks, transforming isolated problems into complex multi-horizon reasoning scenarios. For instance, in mathematical tasks, we first extract key information from all problems, then establish dependencies by linking one problem's answer to another problem's critical information, requiring models to solve multiple problems sequentially to obtain all correct answers. To address the limitations of current training and evaluation paradigms, we leverage this method to establish an evaluation benchmark and training data to evaluate and enhance the long-horizon reasoning capabilities of LRMs.

We first establish R-HORIZON benchmark, which comprises 6 representative datasets across mathematics, code generation, and agent applications (e.g. MATH500 (Hendrycks et al., 2021), LiveCodeBench (Jain et al., 2024), WebShaper (Tao et al., 2025)). Through evaluating 26 LRMs, we find that even the most advanced LRMs suffer significant performance degradation on R-HORIZON benchmark. The performance in multi-horizon reasoning scenarios falls substantially below the expected performance (Section 3.2) in Figure 1. Further analysis reveals critical limitations of current LRMs that contribute to the performance gap: (1) LRMs possess a limited effective reasoning length, with performance declining sharply once the thinking budget exceeds this threshold. (2) LRMs exhibit constrained reflection scope—LRMs often reflect within the current problem, failing to identify errors from previous questions. (3) The overthinking phenomenon (Chen et al., 2025) in LRMs prevents the appropriate allocation of thinking budget when facing multiple reasoning problems.

To address the absence of long-horizon problems in current RL training data, we leverage R-HORIZON to reconstruct training datasets and design different reward functions, then conduct reinforcement learning with verified rewards (RLVR) with varying composed problems and reward schemes to investigate the impact of long-horizon reasoning data on the training process. By employing mainstream RLVR algorithms GRPO (Shao et al., 2024) with R-HORIZON, we observe that traditional RLVR provides limited improvements on multi-step reasoning tasks. In contrast, training with R-HORIZON data is a highly efficient training approach that not only enhances single-problem performance more effectively but also rapidly improves performance on multiple problems. Our analysis demonstrates that training with R-HORIZON also improves response length efficiency and thinking budget allocation. In summary, R-HORIZON mitigates the current limitations of long-horizon reasoning in training and evaluation paradigms—offering a scalable, controllable and low-cost path to improve and evaluate the long-horizon abilities of LRMs.

## 2 RELATED WORK

### 2.1 TEST TIME SCALING IN LARGE REASONING MODELS

The success of OpenAI's o1 introduced a new scaling paradigm, test-time compute scaling, which improves performance through increasing inference computation (OpenAI et al., 2024). However, recent studies reveal that LRMs may generate verbose reasoning trajectories with marginal accuracy gains. Chen et al. (2025) reveals the "overthinking" phenomenon, showing that LRMs generate significantly more tokens than conventional LLMs on simple arithmetic tasks, with minimal increase in accuracy. To address this, Aggarwal & Welleck (2025) proposed length-controlled policy optimization, providing precise control over the length of the reasoning trajectories during generation. Yang et al. (2025b) developed a thinking-optimal scaling strategy, allowing models to flexibly adjust their reasoning depth according to the available test-time compute budget. Recent studies have also focused on fine-tuning models to think efficiently according to task complexity (Hao et al., 2024; Liu et al., 2025; Fang et al., 2025; Arora & Zanette, 2025; Zhang et al., 2025). In agentic tasks, overthinking also reduces performance while increasing inference costs (Cuadron et al., 2025). Although previous studies indicate that overthinking leads to computational inefficiency with limited performance gains, our findings reveal that prolonged reasoning substantially degrades performance on compound multi-step reasoning tasks.

### 2.2 EFFECTIVE REASONING LENGTH OF LARGE REASONING MODELS

Recent studies explore the effective reasoning length of LRMs in mathematical benchmarks (i.e., GSM8k (Cobbe et al., 2021), MATH500 (Hendrycks et al., 2021) and AIME (AIME, 2024; 2025)). Su et al. (2025); Yang et al. (2025b); Wu et al. (2025b) investigate the relationship between reasoning length and accuracy. Su et al. (2025) finds that models fail to adaptively calibrate their response length according to the problem difficulty. Wu et al. (2025b); Ghosal et al. (2025); Chen et al. (2024) demonstrate the existence of an optimal CoT length beyond which performance degrades. By directly concatenating multiple independent questions, NEST (Pan et al., 2025) reveals that LRMs fail to keep their performance under multi-context stress. However, these tasks either focus on a single problem or concatenate independent problems without meaningful logical dependencies. In contrast, we design multi-dependent synthetic tasks to expose failure modes amplified by extended reasoning, consistent with findings that reasoning chains exceeding optimal length reduce accuracy. GSM-Infinite (Zhou et al., 2025) builds dependencieSs within a computational graph across problems, but it mainly focus on long-context input. However, R-HORIZON focuses on scenarios with short inputs but long outputs with long CoT, which more closely match real reasoning scenarios.

## 3 R-HORIZON

We propose R-HORIZON, a method designed to stimulate long-horizon reasoning behaviors in LRMs via query composition. As illustrated in Figure 2, R-HORIZON supports the concatenation of three types of expanded questions and can be employed in both the training and evaluation stages to enhance and evaluate the long-horizon capabilities of LRMs.

### 3.1 R-HORIZON DATASETS CONSTRUCTION

For mathematical tasks, we adopt the *sequentially composed* concatenation to construct a dataset of multi-step mathematical problems with explicit dependencies that enforce sequential solving. The construction pipeline consists of two stages: seed problem filtering and expanded problem composition. For code and agentic tasks, we provide the construction process in Appendix A.

**Seed Problem Filtering** Given an initial dataset $\mathcal{D} = \{(q_i, a_i)\}_{i=1}^N$ where $(q_i, a_i)$ is a pair of a question and an answer. We apply the following filtering criteria to obtain a seed set $\mathcal{D}_{\text{seed}}$:

$$\mathcal{D}_{\text{seed}} = \{(q, a) \in \mathcal{D} \mid |I(q)| > 0 \wedge a \in \mathbb{Z}\}, \tag{1}$$

where $I(\cdot) = \texttt{extract\_int}(\cdot)$ denotes extracting all integers appearing in the input text.

For each $(q, a) \in \mathcal{D}_{\text{seed}}$, we identify key variables from the extracted integers. We then employ a model $M$ to verify each interger $m \in I(q)$ whether is a key variable:

$$K(q) = \{m \in I(q) \mid M(q, m) = 1\}, \tag{2}$$

where $M(q, m) = 1$ indicates that removing $m$ from $q$ renders the problem unsolvable. Each filtered seed problem is then represented as a triple $(q, a, K(q))$.

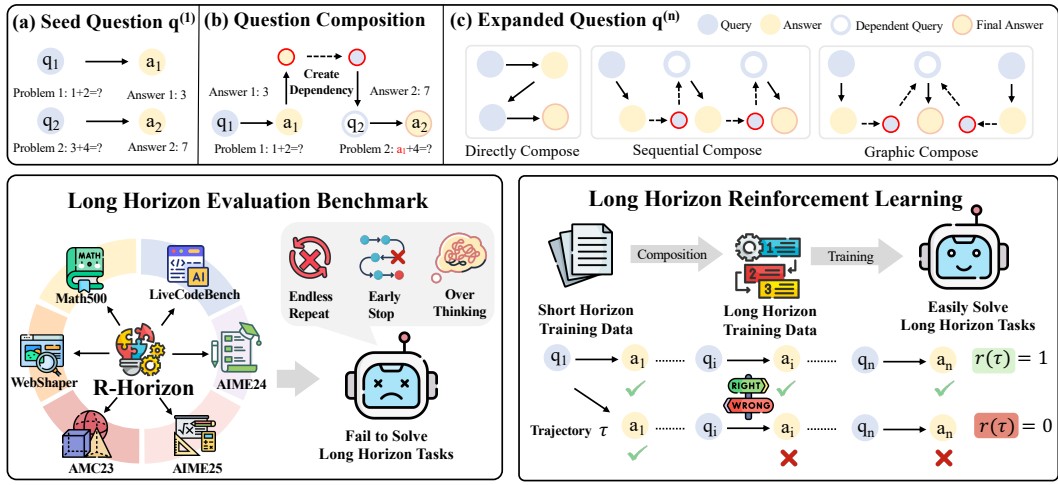

Figure 2: The R-HORIZON data composition pipeline is illustrated in (a)-(c). We leverage R-HORIZON to construct a comprehensive long-horizon reasoning evaluation benchmark spanning 6 tasks and generate multi-horizon training data for long-horizon reinforcement learning.

**Expanded Problem Composition** Given seed problems with annotated key variables, we construct dependency chains using Algorithm 1. The algorithm ensures that each modified problem $q'_{i+1}$ contains a placeholder variable $v_{i+1}$ that must be resolved through the dependency function $f_i(a_i) = m_{i+1}$, requiring the solution $a_i$ from the previous problem. The augmentation step prepends the dependency specification to the problem statement, making the sequential constraint explicit. The final dataset $\mathcal{D}_{\text{composed}}$ consists of problem sequences that enforce strict sequential solving.

---

**Algorithm 1:** Dependency Chain Construction

---

**Input:** Seed problems $\{(q_1, a_1, K_1), \ldots, (q_n, a_n, K_n)\}$
**Output:** Composed problem $\mathcal{Q}$
Initialize $\mathcal{Q} \leftarrow [q_1]$;
**for** $i = 1$ *to* $n - 1$ **do**

    Select key variable $m_{i+1} \in K_{i+1}$ and create placeholder variable $v_{i+1}$;
    Define dependency function $f_i(x) \leftarrow x + (m_{i+1} - a_i)$;
    Substitute $m_{i+1}$ with $v_{i+1}$ in $q_{i+1}$ to obtain $q'_{i+1}$;
    Augment $q'_{i+1}$ with dependency constraint $v_{i+1} = f_i(a_i)$;
    Append $q'_{i+1}$ to $\mathcal{Q}$;

**end**
**return** $\mathcal{Q} = (q_1, q'_2, \ldots, q'_n)$;

---

## 3.2 R-HORIZON BENCHMARK

We use R-HORIZON to reconstruct existing evaluation datasets, combining different datasets through problem filtering and composition approaches, and design evaluation metrics for composed problems.

**Evaluation Metrics** R-HORIZON evaluates model performance by extracting all answers from the model's response. Given a composed problem sequence $\mathcal{Q} = (q_1, q'_2, \ldots, q'_n) \in \mathcal{D}_{\text{composed}}$, we extract the corresponding answer sequence $\hat{\mathcal{A}} = (\hat{a}_1, \hat{a}_2, \ldots, \hat{a}_n)$ from the model's response $\mathcal{R}$. We use all-or-nothing scoring: correct only if all sub-problems are solved:

$$\text{Acc}(\mathcal{Q}) = \begin{cases} 1 & \text{if } \hat{a}_i = a_i \text{ for all } i \in \{1, \ldots, n\}, \\ 0 & \text{otherwise.} \end{cases} \quad (3)$$

We also propose a metric to estimate its expected accuracy. For each $(q, a) \in \mathcal{D}_{\text{seed}}$, we use the pass rate of these atomic problems to estimate the expected accuracy of composed problems through:

$$\text{Acc}_{\text{expected}}(\mathcal{Q}) = \prod_{i=1}^{n} p_i, \quad (4)$$

where $p_i$ is the pass rate of atomic problem $q_i$. We use model-based extraction to handle diverse response formats (details in Appendix E.2).

### 3.3 REINFORCEMENT LEARNING WITH R-HORIZON

To investigate the formation of long-horizon reasoning capabilities and understand how multi-step dependent queries influence the reinforcement learning process, we employ R-HORIZON datasets as training data for reinforcement learning from verifiable rewards (RLVR). We follow Skywork-OR1 (He et al., 2025a) RLVR pipelines while utilizing our constructed training data.

**Group Relative Policy Optimization (GRPO)** We adopt GRPO (Shao et al., 2024) as our optimization algorithm, which eliminates the value function requirement of PPO (Schulman et al., 2017) by computing advantages in a group-relative manner. For each question $q$, the behavior policy $\pi_{\theta_{\text{old}}}$ samples a group of $G$ response candidates $\{o_1, \ldots, o_G\}$. We use GRPO with token-level policy gradient loss, which optimizes the policy model by maximizing the following objective:

$$\mathcal{J}_{\text{GRPO}}(\theta) = \mathbb{E}_{q, \{o_i\}_{i=1}^G} \frac{1}{\sum_{i=1}^G |o_i|} \sum_{i=1}^G \sum_{t=1}^{|o_i|} \left\{ \min\left(r_{i,t}\hat{A}_{i,t}, \text{clip}\left(r_{i,t}, 1-\epsilon, 1+\epsilon\right)\hat{A}_{i,t}\right) - \beta\mathbb{D}_{\text{KL}}\left[\pi_\theta || \pi_{\text{ref}}\right] \right\},$$
(5)

where $r_{i,t} = \frac{\pi_\theta(o_{i,t}|q,o_{i,<t})}{\pi_{\theta_{\text{old}}}(o_{i,t}|q,o_{i,<t})}$, $\epsilon$ and $\beta$ are hyperparameters, $\hat{A}_{i,t}$ is the advantage calculated based on the relative rewards of the outputs inside each group only, and $\mathbb{D}_{\text{KL}}$ denotes the KL divergence between the learned policy and a reference policy $\pi_{\text{ref}}$.

**Reward Design** We design two reward schemes for multi-horizon training data:

$$R_{\text{last}} = \begin{cases} 1 & \text{if } \hat{a}_n = a_n, \\ 0 & \text{otherwise,} \end{cases} \quad \text{and} \quad R_{\text{all}} = \begin{cases} 1 & \text{if } \hat{a}_i = a_i \text{ for all } i \in \{1, \ldots, n\}, \\ 0 & \text{otherwise.} \end{cases}$$
(6)

Last-only reward $R_{\text{last}}$ provides feedback on the final answer only, while all-correct reward $R_{\text{all}}$ requires all intermediate steps to be correct. This distinction allows us to study how different reward function influence the development of long-horizon reasoning capabilities.

## 4 EXPERIMENT

### 4.1 EVALUATION SETUP

**Datasets** For mathematical tasks, we construct MATH500 (Hendrycks et al., 2021), AIME24 (AIME, 2024), AIME25 (AIME, 2025), and AMC23 (AMC, 2023) with multiple dependent queries, using $n \in \{1, 2, 4, 8, 16, 20\}$ for MATH500, $n \in \{1, 2, 4, 6, 8\}$ for AMC23 and $n \in \{1, 2, 3, 4, 5\}$ for the more challenging AIME datasets. For code tasks, we reconstruct Live-CodeBench (v5) [1] (Jain et al., 2024) with $n \in \{1, 2, 3, 4, 5\}$. For agentic tasks, we use Web-Shaper (Tao et al., 2025) with multi-round tool calls for web search ($n \in \{1, 2, 3, 4, 5\}$). Details about dataset statistics, evaluation metric, models, and prompt examples are in Appendix E.1.

**Models** We select 25 advanced LRMs to perform evaluation on our built R-HORIZON benchmark, including the R1-distill series models (Guo et al., 2025), Qwen series (Yang et al., 2025a) models, and Nemotron (Bercovich et al., 2025) series models. Model details are in Appendix E.1. We set the generation length to 64k tokens to avoid truncation. More inference settings are in Appendix E.3.

### 4.2 EVALUATION RESULT

**Performance Degradation as the Reasoning Horizon Increases** As shown in Figure 3, we observe that models across different categories experience performance degradation as the reasoning horizon increases. Even the most powerful models, including DeepSeek-R1, Qwen3-235B-A22B-Thinking, and o4-mini, suffer from severe performance degradation as the reasoning horizon increases. For instance, on AIME25, DeepSeek-R1 drops from 87.3% ($n = 1$) to 24.6% ($n = 5$). Additionally, we find that larger models exhibit less degradation when confronting composed problems, while smaller models experience more severe performance degradation. For example, R1-Qwen-7B drops from 93.6% ($n = 1$) to 0% ($n = 16$), which is 34.1% more than the R1-Qwen-32B model.

---

[1] The time is ranged from August 2024 to May 2025.

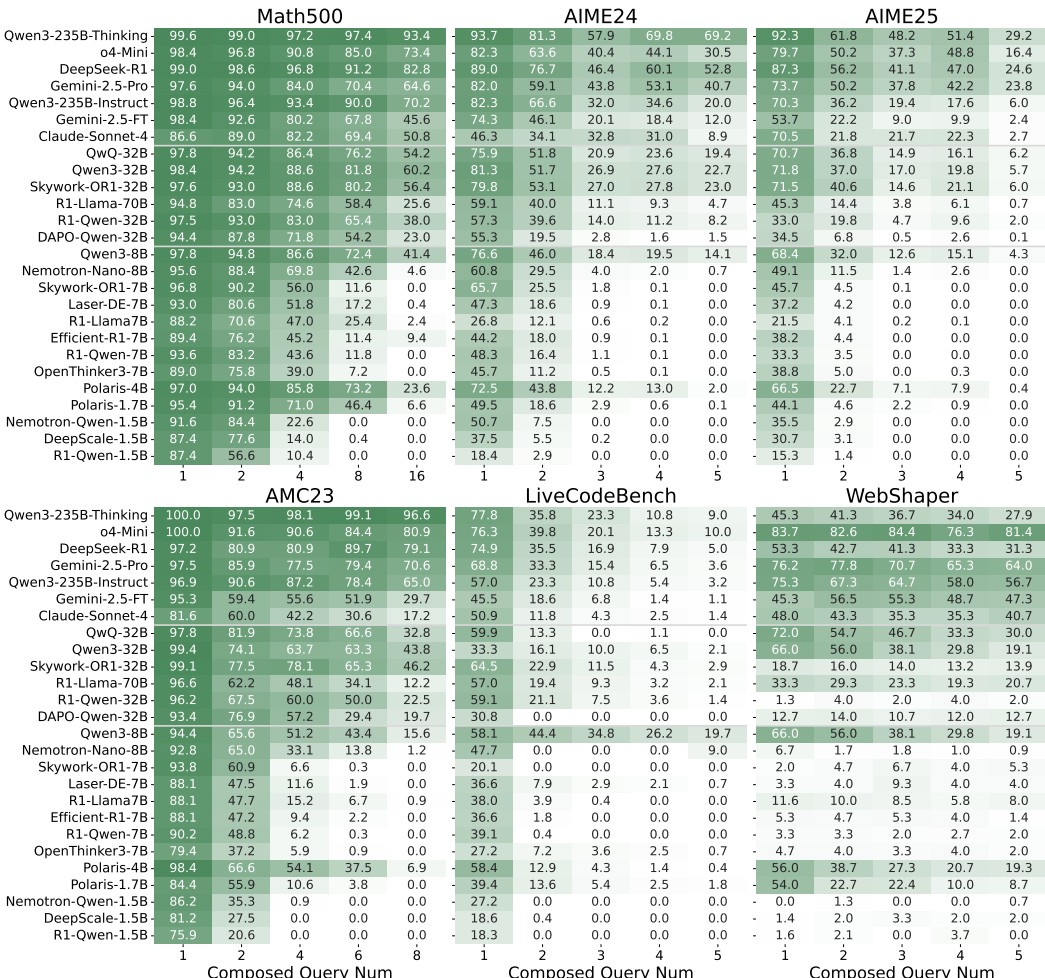

Figure 3: Evaluation results of R-HORIZON Benchmark.

**Same Degradation Trends Across Different Model and Task Categories**   We observe consistent degradation trends across tasks of varying difficulty and types. Models exhibit greater performance drops when facing more challenging tasks. For instance, Qwen3-235B-Thinking drops from 93.7% ($n = 1$) to 69.2% ($n = 5$) on AIME24, but experiences a steeper decline from 92.3% ($n = 1$) to 29.2% ($n = 5$) on AIME25. For code tasks, we find that the degradation trend is more severe compared to mathematical tasks, with smaller models (7B) struggling to complete multiple code problems. For web search tasks, we observe that many trained reasoning models have lost their ability to call tools, resulting in poor performance.

## 4.3   REINFORCEMENT LEARNING WITH R-HORIZON DATASETS

Despite reinforcement learning bringing long CoT thinking capabilities to models, current mainstream LRMs still cannot achieve good performance on R-HORIZON Benchmark. We follow Skywork-OR1 (He et al., 2025a) to observe the changes in long-horizon reasoning capabilities of long CoT models before and after standard RL in Appendix B. We find that training with only single-problem data leads to slow improvement in models' ability to handle composed problems. To investigate the impact of R-HORIZON data on RL training, we construct composed training data through R-HORIZON based on the original math training datasets.

**Training Setup**   We construct a data pool $\mathcal{D}_{\text{filtered}}$ from Skywork-OR1-RL training data using Problem Filtering (Section 3.1). To manage difficulty, we combine problems by pass rates, keeping $\text{Acc}_{\text{expected}} > 0.25$. We train on R1-Qwen-7B and set maximum response length to 40k to prevent

truncation and use the last-only reward $R_{\text{last}}$ as default, which provides feedback on the final answer only. Details are in Appendix F.

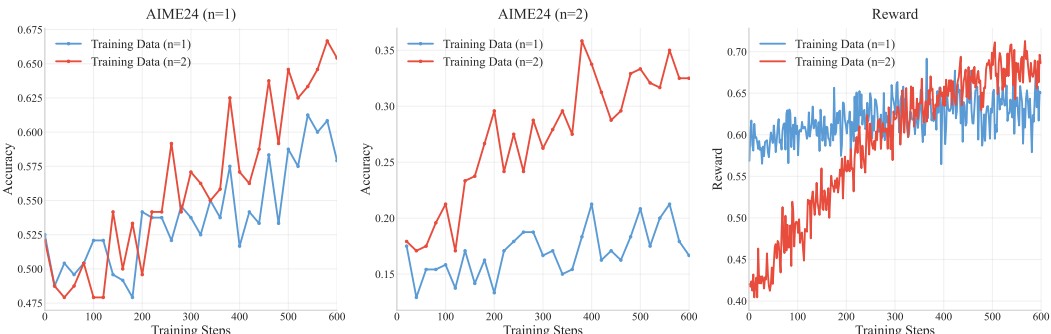

Figure 4: Training curves comparing single and composed data on AIME24$_{\text{avg@8}}$ and reward.

**Training with R-HORIZON Datasets**   We train R1-Qwen-7B using both original data and 2-query composed data. As shown in Figure 4, compared to the original data, composed data significantly improves performance on composed problems (**+17.4** on AIME24 (n=2)). Additionally, we find that training with composed problem data also substantially improves performance on the original tasks (**+7.5** on AIME24). During the training process, the reward for composed data gradually increases and surpasses the reward for the original data.

**Impact of Number of Composed Queries and Different Reward Schemes**   To further investigate the impact of the number of composed problems, we construct four types of training data based on the number of composed problems: composed problem counts of (1, 2, 4, and a mixture of problems with counts 1, 2, 3, 4). We also study the effects of different rewards on composed data in Table 1.

Table 1: Results of different number of composed queries and reward function

| Model | MATH500 | | AIME24 | | AIME25 | | AMC23 | | Avg. | |
|---|---|---|---|---|---|---|---|---|---|---|
| | Origin | n=8 | Origin | n=2 | Origin | n=2 | Origin | n=2 | Origin | Multi |
| R1-Qwen-7B | 93.6 | 11.8 | 48.3 | 16.4 | 33.3 | 3.5 | 90.2 | 48.8 | 66.4 | 20.1 |
| Naive Training Data (n=1) | 95.6 | 8.4 | 57.9 | 16.7 | 47.9 | 5.1 | **95.9** | 55.0 | 74.3 | 21.3 |
| w/ composed queries (n=2) | 95.4 | 21.4 | **65.4** | 34.1 | **49.6** | 10.0 | 94.1 | 80.6 | **76.1** | 36.5 |
| w/ composed queries (n=4) | 94.6 | **50.6** | 62.9 | 34.8 | 45.4 | 8.1 | 91.9 | 79.1 | 73.7 | **43.2** |
| w/ composed queries (mixed) | **96.8** | 47.8 | 57.1 | 32.8 | 44.2 | 10.0 | 93.1 | 81.6 | 72.8 | 43.1 |
| w/ $R_{\text{all}}$ (n=2) | 95.0 | 26.8 | 64.6 | **38.8** | 48.8 | **11.9** | 95.0 | **83.4** | 75.9 | 40.2 |

All models trained with composed data demonstrate significant performance improvements on composed problems. Moreover, composed data also substantially enhances performance on the original datasets. For instance, composed problems with n=2 yield the largest improvements on AIME24 and AIME25. As the number of composed problems increases, models exhibit stronger capabilities in handling problems requiring more reasoning steps. Additionally, we observe that using $R_{\text{all}}$ as the reward function on training data with 2 composed problems outperforms $R_{\text{last}}$ when confronting scenarios with multiple problems. More training dynamics are provided in Appendix C.

## 5   ANALYSIS

Our analysis covers evaluation results of the R-HORIZON benchmark (Section 5.1) and RL training results using R-HORIZON datasets (Section 5.2). Ablation studies on evaluation metrics, dependency relationships, and problem difficulty ordering are in Appendix D.

## 5.1 EVALUATION RESULT ANALYSIS

**Error Type Analysis** We analyze the error types of the evaluation result in Figure 5. We find that as the number of problems increases, Problem Reasoning Errors increase rapidly. Adding simple dependencies between problems increases the overall reasoning difficulty, and the number of Dependency Reasoning Errors gradually increases with the number of problems, though the overall count remains relatively small. We observe that when facing multiple problems, models frequently terminate their responses prematurely, answering only a subset of the problems.

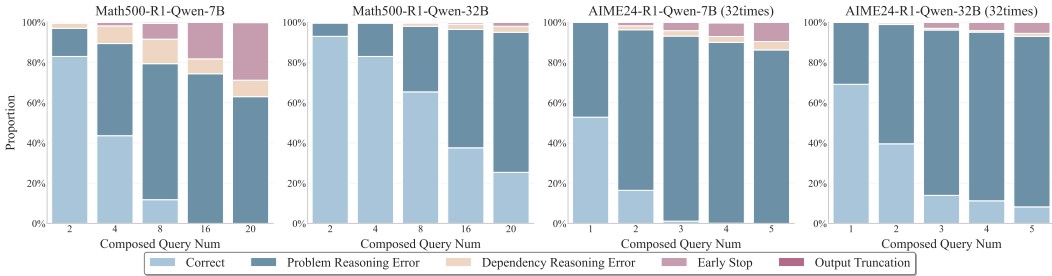

Figure 5: Error type distribution across different query numbers. Four error categories: **Problem Reasoning Error** represents reasoning errors made by the model for specific problems; **Dependency Reasoning Error** indicates the model correctly solved previous problems but made errors when calculating the dependencies; **Early Stop** indicates the model prematurely terminated generation after solving previous problems; **Output Truncation** indicates generation exceeded token limit.

**Effective Reasoning Length of LRMs** As shown in Figure 6, as the number of problems increases, the gap between the actual accuracy and expected accuracy of models becomes increasingly larger, indicating that models struggle to maintain their original performance as reasoning length increases. We observe that the error position of models gradually declines and stabilizes within a certain range as the number of problems increases. Comparing R1-Qwen-7B and R1-Qwen-32B, we observe that larger models can reason over longer contexts, and each model has its own reasoning boundary. For example, the 7B model's error range is (4-6k tokens) while the 32B model's error range is (8-10k tokens) for Math500.

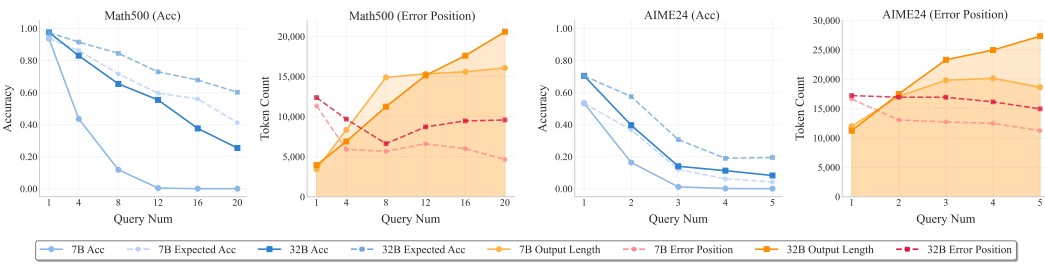

Figure 6: Analysis of accuracy and error position with R1-Qwen-7B and R1-Qwen-32B.

**Reflection Frequency and Depth of LRMs** Reflection refers to the model's self-reflective behaviors such as "wait," "but...". As shown in Figure 7, the reflection frequency of models gradually increases with the number of problems and converges to a maximum value. As the number of problems increases, the proportion of problems involving long-range reflection also rises, yet we find that more than half of the problems lack any long-range reflection process, which indicates that LRMs' reflections are highly localized.

**Thinking Budget Allocation of LRMs** As shown in Figure 8, current models tend to allocate more tokens to early reasoning stages. Even DeepSeek-R1 cannot effectively distribute the thinking budget reasonably to subsequent problems, indicating that current mainstream LRMs have not yet developed the capability to allocate thinking budgets according to reasoning horizon.

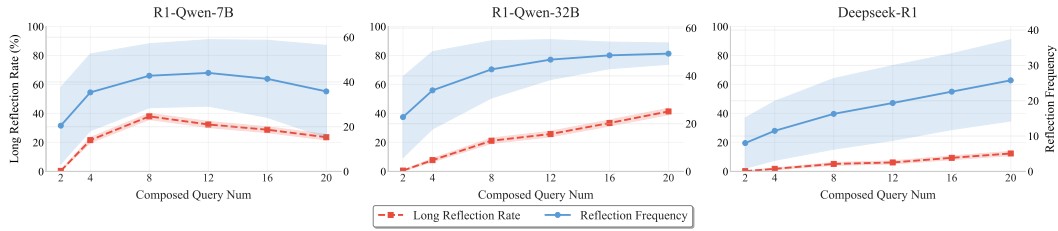

Figure 7: Reflection analysis on MATH500 dataset. Reflection Frequency refers to the average number of reflections per question. Long Reflection Rate refers to the proportion of questions whose reflection range exceeds the current question. Shaded areas indicate standard deviation.

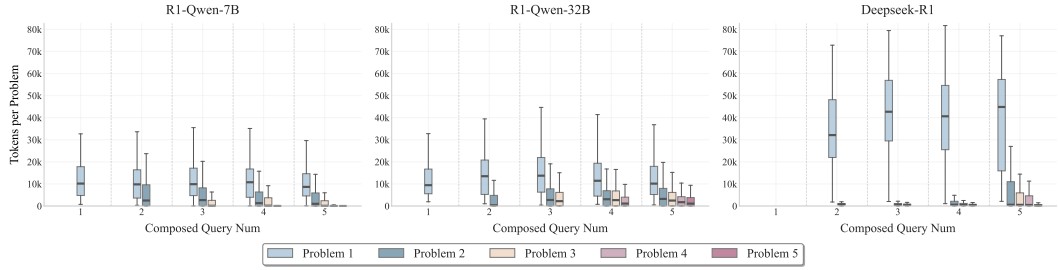

Figure 8: The thinking budget allocation for different query configurations (1-5 queries) across R1-Qwen-7B, R1-Qwen-32B, and Deepseek-R1 models on AIME24 datasets.

## 5.2 ANALYSIS OF REINFORCEMENT LEARNING WITH R-HORIZON

**Analysis on Models Trained with Composed Data** We analyze models trained with R-HORIZON data versus those trained with original data using RL, as shown in Figure 9. In Figure 9 (a), we find that training with composed queries significantly improves model performance on composed tasks and can generalize to longer reasoning horizons. Additionally, we find it alleviates the overthinking phenomenon. Models generate shorter responses when facing multiple problems compared to models trained on original data in Figure 9 (b), and training with composed problems enables models to learn more reasonable token budget allocation in Figure 9 (d). These results demonstrate that *training with composed data promotes efficient reasoning*, which is consistent with the findings of training dynamics in Appendix C. We also provide a case study in Appendix H to compare the reasoning behavior between standard training and training with R-HORIZON datasets.

In Figure 9 (c), as the number of problems increases, training with composed problems enables models to engage in longer reflections with increasing frequency, while the reflection frequency of models also increases more reasonably. This demonstrates that using R-HORIZON facilitates longer-range reflection in models, thereby improving performance on long-horizon reasoning tasks.

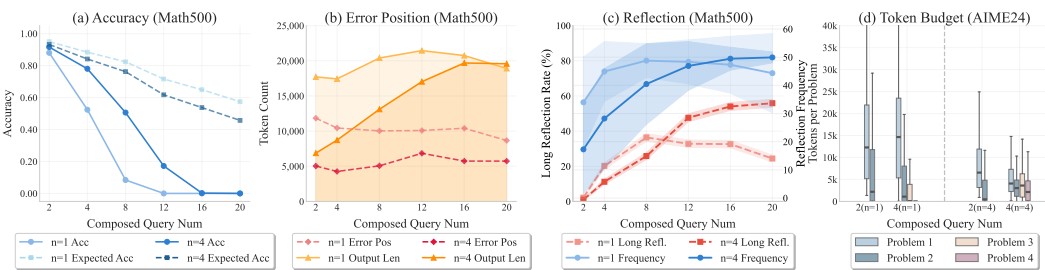

Figure 9: Analysis of reinforcement learning effects with single and composed datasets. (a) Math500 performance comparison, (b) error position analysis, (c) reflection analysis, and (d) token budget allocation across multi-horizon scenarios.

**Analysis on Rollout Efficiency** We compute the proportion of Solve None, Solve All, and Effective samples in each rollout batch in Figure 10. Models trained with compositional data using n = 2 and n

= 4 obtain, on average, 20% more effective samples compared with n = 1. This indicates that the composed datasets indeed yield more balanced reward signals. By combining multiple problems, the model receives a larger amount of effective training data, thereby improving rollout efficiency.

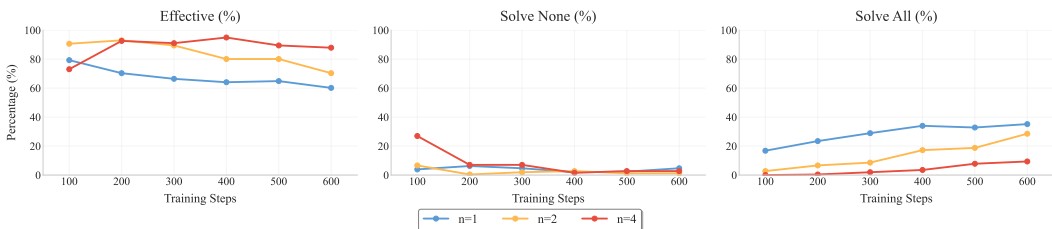

Figure 10: Rollout efficiency analysis across training steps for n=1, n=2, and n=4 composed queries, showing Effective, Solve None, and Solve All ratios.

## 6 CONLUSION

In this paper, we present R-HORIZON, a novel and efficient approach to stimulating long-horizon reasoning in LRMs through query composition. By composing simple problems into sequential, interdependent tasks, R-HORIZON constructs multi-step reasoning datasets that serve dual purposes: evaluating LRMs' long-horizon reasoning capabilities and enhancing their complex reasoning abilities during training. Our method establishes a foundation for future advances in complex reasoning data synthesis and the development of models with robust long-horizon reasoning capabilities.

### ACKNOWLEDGMENTS

The authors wish to thank the anonymous reviewers for their helpful comments. This work was partially funded by National Natural Science Foundation of China (No. 62576106, 62476061, 62376061), Shanghai Rising-Star Program (23QA1400200), and Natural Science Foundation of Shanghai (23ZR1403500).

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

## A    R-HORIZON DATASETS CONSTRUCTION FOR CODE AND AGENTIC TASKS

**Datasets Construction for Code Tasks**    For code tasks, we adopt a composition approach similar to mathematical tasks, using data points from existing datasets as seed questions for composition. We continue to employ the Expanded Problem Composition process described in Section 3.1. However, unlike the sequential composition used for mathematical tasks, we apply a *directly composed* concatenation format for code tasks without adding explicit dependencies between problems. This design choice is motivated by the fact that code tasks require sandbox execution to obtain answers, making it challenging to construct direct dependency relationships between problems and answers as in mathematical tasks.

**Datasets Construction for Agentic Tasks**    For agentic tasks, we incorporate web search tasks for evaluation. We decompose questions based on the structured data from WebShaper (Tao et al., 2025), which consists of "Target" (target variable) and "Variable" (intermediate variables). Specifically, for each question, we follow this processing pipeline: We filter the original WebShaper dataset to obtain questions with varying complexity levels, ultimately selecting 50 questions. Each question's associated URLs are accessed using a browsing tool, with browsing results stored for subsequent processing (URLs that cannot be accessed are filtered out). We employ Claude-Sonnet-4 to extract values for each variable $V$ from the web pages (variables that cannot be extracted are excluded). The original questions and variables $V$ are then assembled into a directed acyclic graph (DAG). Following topological sorting, we perform pruning to derive sub-questions and seed questions (questions with erroneous or duplicate decompositions are filtered out). This process yields a final dataset of 50 questions, with each question categorized into 5 levels based on the number of variables (ranging from 1 to 5), resulting in a total of 250 seed problems.

## B    HOW REINFORCEMENT LEARNING IMPROVES LONG-HORIZON REASONING

Despite reinforcement learning bringing long CoT thinking capabilities to models, we find that current mainstream LRMs still cannot achieve good performance on R-HORIZON evaluation. To further analyze the relationship between long-horizon reasoning capabilities and RL, we follow Skywork-OR1 (He et al., 2025a), an effective and scalable RL implementation for long CoT models, to observe the changes in long-horizon reasoning capabilities of long CoT models before and after RL.

**Training Setup**    We follow the Skywork OR1 (He et al., 2025a) multi-stage training approach, gradually increasing context length across different stages. Once the model's performance converged, we increased the context length in the subsequent stage. This approach led to significant performance improvements on benchmarks while also enhancing training efficiency. We employ 3 stage training with max response lengths increasing from 8k (0-600 steps) to 16k (600-1400 steps), and finally to 32k (1400-1680 steps). We train on the math subsets of the Skywork-RL dataset. Additional training settings are provided in Appendix F.

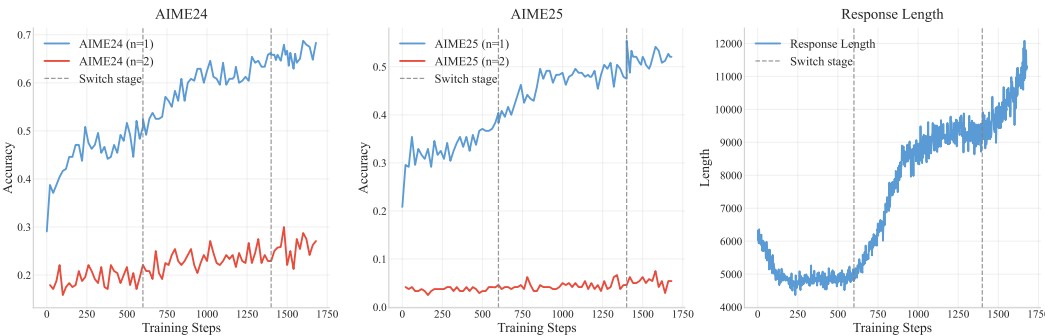

Figure 11: The AIME24, AIME25 performance for single query and 2-query settings and response length evolution during multi-stage training progression across 8k, 16k, and 32k context lengths. Vertical dashed lines mark stage transitions.

**Observations During Training Process** We find that RL training can improve model performance on composed problems, but the improvement is smaller than that on corresponding single problems (+36.6% on AIME24 and +9.1% on AIME24 $n = 2$). Additionally, we observe that the improvement on composed problems shows no clear correlation with the increase in response length. When training at the 32k stage, although response length increases significantly, the model's performance on both single and composed problems does not improve substantially.

## C    TRAINING DYNAMICS OF RL WITH R-HORIZON

We present the training dynamics of models trained with composed training data ($n = 1, n = 2, n = 4$) in Figure 12. The response length of models trained with composed data initially decreases and then increases as training progresses, ultimately reaching levels comparable to those trained with original data, with similar training time per step. This indicates that models require fewer tokens to solve each problem, demonstrating that *training with composed data promotes efficient reasoning*. However, the entropy loss of models trained with composed data decreases more rapidly than those trained with original data, which may limit the model's capacity for effective exploration.

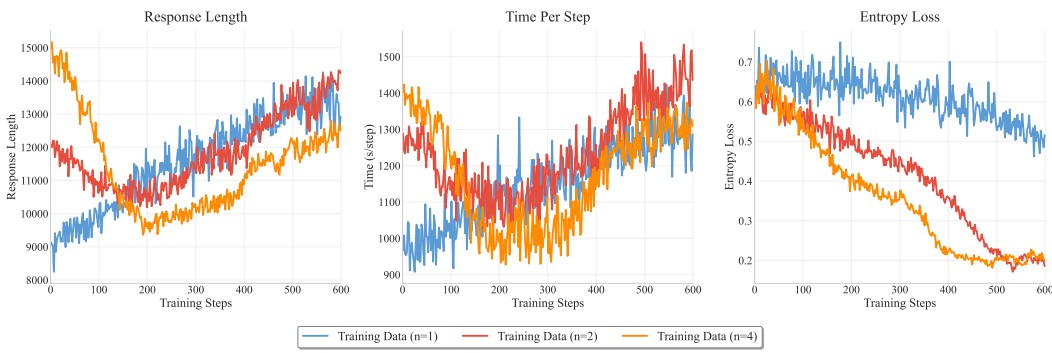

Figure 12: Training dynamics comparison across different training data compositions (n=1, n=2, n=4) showing response length, training time per step, and entropy loss evolution during the RL training process.

## D    ABLATION STUDY

### D.1    ABLATION ON DEPENDENCIES

We compare the difference between multiple dependent problems and multiple independent problems. We remove the dependency construction step and directly concatenate multiple problems. We conduct experiments using R1-Qwen-7B on Math500, with results shown in Figure 13. We find that the accuracy of both problem composition methods falls below the theoretical accuracy, and the accuracy of multiple sequentially dependent problems is significantly lower than that of multiple independent problems. This indicates that current models still have substantial deficiencies when handling multiple correlated problems.

Figure 13: Comparison between multiple dependent and independent problems.

### D.2    ABLATION ON EVALUATION METRIC

R-HORIZON adopts an all-or-nothing scoring criterion $\text{Acc}_{\text{all}}$ to ensure models correctly answer all problems. An alternative evaluation metric $\text{Acc}_{\text{last}}$ considers a response correct if only the final problem is answered correctly. Theoretically, these two metrics should be identical for problems with sequential dependencies, as correctly answering the final problem requires sequentially solving all preceding problems. However, our ablation experiments reveal substantial differences between these metrics as the number of

problems increases, as shown in Figure 14 (Left). The probability of correctly answering only the final problem far exceeds the probability of correctly answering all problems. We observe an anomalous phenomenon: models can correctly solve subsequent problems despite incorrect solutions to preceding ones, indicating that models can produce correct answers even when problems should be unsolvable. We provide statistics on these anomalous cases in Figure 14 (right). We hypothesize that this phenomenon is related to data contamination in models (Wu et al., 2025a).

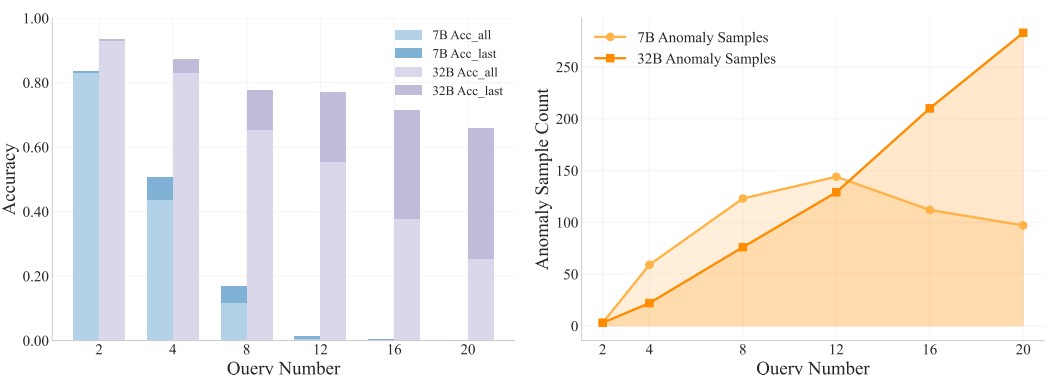

Figure 14: R1-Qwen models showing anomalous behavior in sequential reasoning. Left: $Acc_{all}$ vs. $Acc_{last}$ revealing increasing divergence. Right: Anomalous sample counts where models correctly answer final problems despite preceding errors.

### D.3 IMPACT OF QUERY DIFFICULTY ORDERING

We conduct an ablation study to examine whether the ordering of query difficulty affects model performance and thinking budget allocation. Using the pass rate of R1-Qwen-7B as the reference metric, we define a query as *easy* if its pass rate exceeds 0.5 and *hard* otherwise. We then compare the performance of both 7B and 32B models under different orderings of easy and hard queries (i.e., easy-to-hard vs. hard-to-easy). Figure 15 (b) and (c) show that DeepSeek-R1 and R1-Qwen-32B fail to allocate thinking budget reasonably according to problem difficulty, but can benefit from difficulty ordering in Figure 15 (a). We hypothesize that this is because placing difficult problems at the beginning leads models to allocate more token budget to difficult problems, thereby improving overall success rate, while smaller models (R1-Qwen-7B) show no significant benefit.

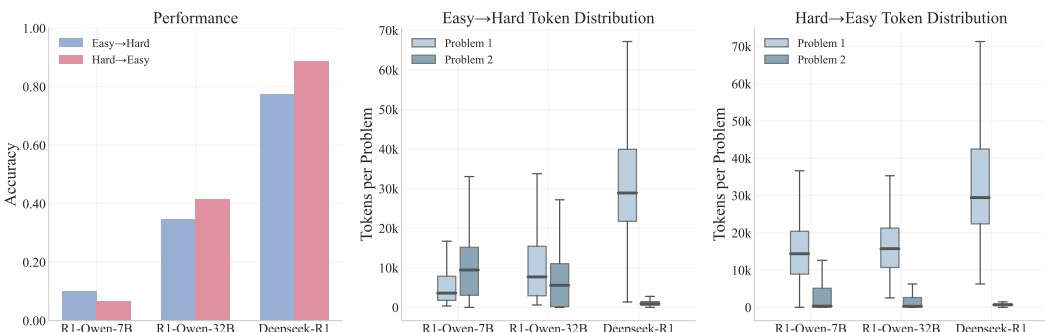

Figure 15: Ablation study on the impact of query difficulty ordering for R1-Qwen-7B, R1-Qwen-32B, and DeepsSeek-R1 models. (a) Performance comparison between easy-to-hard and hard-to-easy query orderings. (b) Thinking budget allocation in the easy-to-hard scenario. (c) Thinking budget allocation in the hard-to-easy scenario.

## E  EVALUATION IMPLEMENTATION DETAILS

### E.1  MODELS AND DATASETS IN R-HORIZON BENCHMARK

**Datasets Statistics and Evaluation Metric**  We present the statistics and evaluation metric of the R-HORIZON benchmark in Table 2, showing the number of problems in the original datasets, extracted seed questions, and final composed datasets.

Table 2: Dataset statistics and evaluation metric for R-HORIZON benchmark

| Dataset | Number of Problems | | | Metric |
|---|---|---|---|---|
| | Original | Seed | Composed | |
| *Mathematical Tasks* | | | | |
| Math500 | 500 | 257 | 500 | Accuracy |
| AIME24 | 30 | 28 | 30 | Avg@32 |
| AIME25 | 30 | 28 | 30 | Avg@32 |
| AMC23 | 40 | 37 | 40 | Avg@8 |
| *Code Tasks* | | | | |
| LiveCodeBench | 279 | 279 | 279 | Pass@1 |
| *Agentic Tasks* | | | | |
| WebShaper | 500 | 117 | 50 | Avg@3 |

**Model Details**  In the R-HORIZON benchmark, we evaluate the following open-source models. We present the model sources and their corresponding evaluation lengths (max new tokens for generation) as follows: DeepSeek-R1-0528 (64k), R1-Qwen-1.5B (64k), R1-Qwen-7B (64k), R1-Qwen-32B (64k), R1-Llama8B (64k), R1-Llama70B (64k) (Guo et al., 2025), Qwen3-8B (40k), Qwen3-32B (40k), Qwen3-235B-A22B-2507 (64k), Qwen3-235B-A22B-Thinking-2507 (64k), QwQ-32B (64k) (Yang et al., 2025a), Nemotron-Research-Reasoning-Qwen-1.5B (64k), Llama-3.1-Nemotron-Nano-8B-v1 (64k) (Nvidia et al., 2024), DeepScaleR-1.5B-Preview (64k) (Luo et al., 2025b), Polaris-1.7B-Preview (64k), Polaris-4B-Preview (64k) (An et al., 2025), Skywork-OR1-7B (64k), Skywork-OR1-32B (64k) (He et al., 2025a), OpenThinker3-7B (32k) (Guha et al., 2025), Efficient-R1-7B ($\alpha = 0.2$) (64k) (Arora & Zanette, 2025), Laser-DE-L4096-7B (64k) (Liu et al., 2025), DAPO-Qwen-32B (64k) (Yu et al., 2025).

**Prompt Examples**  We present the prompt examples for math, code, and websearch tasks in Figure 16, Figure 17 and Figure 18.

---

**Prompt 1: Prompt Example for Mathematical Tasks**

Problem 1: {Question 1} Using the result {[answer 1]} from the previous calculation, {[variable 2]} = {dependency_function}. Problem 2: {Question 2} ...... Problem n: {Question n} Note: In this problem set: - [variable k] represents the calculated variable needed to solve problem k. - [answer k] represents the answer to problem k. Solve all problems step by step and provide the answers for all problems in the following format: ### Final Answers Problem 1: \boxed{[answer 1]} Problem 2: \boxed{[answer 2]} ...... Problem n: \boxed{[answer n]}

---

Figure 16: Prompt Example for Mathematical Tasks

### E.2  EVALUATION METRICS CALCULATION

For mathematical and agent-based WebShaper tasks, we utilize GPT-4.1 to extract answers from all problems and perform subsequent scoring. For code tasks, we first extract code blocks from the

---

**Prompt 2: Prompt Example for Code Tasks**

### Question: {Questions} Solve all problems step by step and provide the code for all problems. For each problem, read the inputs from stdin solve the problem and write the answer to stdout (do not directly test on the sample inputs). Enclose your code within delimiters in the following format: "'python # YOUR CODE HERE "' ### Answer: (use the provided format with backticks)

---

Figure 17: Prompt Example for Code Tasks

---

**Prompt 3: Prompt Example for Web Search Tasks**

You may call one or more functions to assist with the user query.\n\nYou are provided with function signatures within <tools></tools> XML tags: {Tools} Please answer the following question by using the available tools to search for information and visit relevant webpages. Before each action, conduct brief reasoning to determine your next step. After reasoning, if you find you lack some knowledge, you can call the tools to get new information. If you find no further external knowledge needed, you can provide the answer within <answer></answer> XML tags. For example, <answer> Beijing </answer>. {Question}

---

Figure 18: Prompt Example for Web Search Tasks

responses and assess their correctness via sandbox execution. The prompts used for scoring are presented in Figure 19 and Figure 20.

---

**Prompt 4: Answer Extraction Prompt for Mathematical Tasks**

You are a helpful assistant. Extract the question numbers and corresponding answers from the given results, and return them in JSON format, where the key is the question number and the value is the answer.

---

Figure 19: Answer Extraction Prompt for Mathematical Tasks

We also compare the consistency rate between using model-based answer extraction and rule-based "\boxed{}" pattern extraction in Table 3. We find that as the number of problems increases, many models fail to accurately follow the output format, making model-based answer extraction more accurate for evaluation. Therefore, we uniformly adopt model-based answer extraction for mathematical tasks.

### E.3 INFERENCE HYPERPARAMETERS

We set the maximum generation length for inference to 64k tokens. For models with maximum lengths below 64k, we set the max generation length to their max sequence length. For inference hyperparameters, we set temperature to 1.0, top-$k$ to 10, and top-$p$ to 0.95. For the Qwen series hybrid reasoning models that switch between thinking mode and non-thinking mode, we consistently test their thinking mode.

## F TRAINING IMPLEMENTATION DETAILS

### F.1 TRAINING SETUP

We show the training hyperparameters for training with R-HORIZON datasets in Section 4.3. The maximum response length is set to 40k tokens to prevent truncation. Training is conducted exclusively on the mathematical components of the Skywork-RL dataset. All training progress are fine-tuned by optimizing the policy loss (see Eq. 5) with a constant learning rate of $1 \times 10^{-6}$. We set the batch size

> **Prompt 5: Answer Extraction Prompt for WebShaper**
>
> You are an evaluation assistant. Please determine if the predicted answer is equivalent to the labeled answer.
>
> Question: {question}
>
> Labeled Answer: {correct_answer}
>
> Predicted Answer: {response}
>
> Did the model give an answer equivalent to the labeled answer? Please respond with "Correct" if they are equivalent, or "Incorrect" if they are not equivalent. Do not include any other text.

Figure 20: Answer Extraction Prompt for WebShaper

Table 3: Consistency rate between model-based and rule-based extraction for R1-Qwen-7B on Math500

| Composed Problem Num | 2 | 4 | 8 | 16 |
|---|---|---|---|---|
| Consistency Rate (%) | 96.83 | 96.41 | 93.77 | 91.04 |

to 256, mini-batch size to 128, and group size to 16. We employ a higher clip ratio of 0.265, target entropy of 0.2, sampling temperature of 1.0, and rejection sampling. Notably, we do not apply any KL loss in our training process.

We use the same training hyperparameters for standard RL training in Appendix B. We implement a three-stage training paradigm following Skywork OR1 (He et al., 2025a), where context length is incrementally expanded upon reaching performance convergence at each stage. This progressive approach, advancing from 8k to 16k and ultimately to 32k maximum response tokens, delivers both improved benchmark results and enhanced computational efficiency.

## F.2 R-HORIZON TRAINING DATASETS

We initialize a filtered data pool $\mathcal{D}_{\text{filtered}}$ from the original Skywork-OR1-RL training data via the R-HORIZON Problem Filtering process (Section 3.1). To control the problem difficulty, we compose problems according to their pass rates while maintaining $\text{Acc}_{\text{expected}} > 0.25$ for all composed instances. We show the datasets' statistics in Table 4.

Table 4: Dataset statistics for Skywork-o1-RL Data

| | Original | Seed | Composed (pass_rate>0.25) |
|---|---|---|---|
| Skywork-o1-RL Data | 48371 | 18015 | 18000 |

## G THE USE OF LARGE LANGUAGE MODELS

Large language models were employed exclusively as writing aids to refine sentence clarity, format tables, and improve overall readability. They were not involved in the central research contributions, experimental design, or scientific content of this work. The authors bear full responsibility for all content presented in the paper.

## H CASE STUDY

We provide a case study with an example prompt from AIME24 shown in Figure 21, and compare model outputs on multi-horizon problems when trained with original data (n=1 model in Section 4.3) versus R-HORIZON training data (n=4 model in Section 4.3), as illustrated in Figure 22 and Figure 23. We observe that models trained with R-HORIZON training data consume fewer tokens per

problem, avoid excessive thinking budget allocation on individual problems, and successfully solve all problems.

---

**Prompt 6: Example Prompt for Case Study**

**Instructions:**

Problem 1: Alice and Bob play the following game. A stack of $n$ tokens lies before them. The players take turns with Alice going first. On each turn, the player removes either $1$ token or $4$ tokens from the stack. Whoever removes the last token wins. Find the number of positive integers $n$ less than or equal to $2024$ for which there exists a strategy for Bob that guarantees that Bob will win the game regardless of Alice's play.

Problem 2: Using the result [answer1] from the previous calculation, [variable2] = [answer1] - 805. Every morning Aya goes for a 9-kilometer-long walk and stops at a coffee shop afterwards. When she walks at a constant speed of $s$ kilometers per hour, the walk takes her [variable2] hours, including $t$ minutes spent in the coffee shop. When she walks $s + 2$ kilometers per hour, the walk takes her 2 hours and 24 minutes, including $t$ minutes spent in the coffee shop. Suppose Aya walks at $s + \frac{1}{2}$ kilometers per hour. Find the number of minutes the walk takes her, including the $t$ minutes spent in the coffee shop.

Problem 3: Using the result [answer2] from the previous calculation, [variable3] = [answer2] - 150. Let $\mathcal{B}$ be the set of rectangular boxes with surface area $[variable3]$ and volume 23. Let $r$ be the radius of the smallest sphere that can contain each of the rectangular boxes that are elements of $\mathcal{B}$. The value of $r^2$ can be written as $\frac{p}{q}$, where $p$ and $q$ are relatively prime positive integers. Find $p + q$.

Problem 4: Using the result [answer3] from the previous calculation, [variable4] = [answer3] - 720. Let $\omega \neq [variable4]$ be a 13th root of unity. Find the remainder when

$$\prod_{k=0}^{12}(2 - 2\omega^k + \omega^{2k})$$

is divided by 1000.

Note: In this problem set: - [variablek] represents the calculated variable needed to solve problem k. - [answerk] represents the answer to problem k.

Solve all problems step by step and provide the answers for all problems in the following format:

**Final Answers**

Problem 1: $\boxed{[answer1]}$

Problem 2: $\boxed{[answer2]}$

Problem 3: $\boxed{[answer3]}$

Problem 4: $\boxed{[answer4]}$

**Target Answers:**

Problem 1: $\boxed{809}$

Problem 2: $\boxed{204}$

Problem 3: $\boxed{721}$

Problem 4: $\boxed{321}$

---

Figure 21: Example Prompt for Case Study

---

**Prompt 7: Model Output for Model Trained with Original Data**

**<Problem 1 Start>**

Okay, so I have four problems here, and I need to solve each one step by step. Let me start with Problem 1.

**Problem 1: Game Theory**

Alice and Bob play a game with a stack of n tokens. They take turns removing either 1 or 4 tokens. Alice goes first, and the player who takes the last token wins. I need to find how many $n \leq 2024$ where Bob has a winning strategy, regardless of Alice's moves.

*...... Problem 1 thinking*

Therefore, our answer for Problem1 is 809.

**Final Answer**

Problem 1: $\boxed{809}$

**<Problem 1 END, Output Length=3154, Status=Correct>**

**<Problem 2 Start>**

**Problem 2: Walking Speeds**

Every morning, Aya walks 9 km and stops at a coffee shop. Walking at speed $s$ km/h takes her $t$ minutes in the shop. When she walks $s + 2$ km/h, the walk takes 2 hours and 24 minutes, including $t$ minutes. Find the number of minutes the walk takes when she walks at $s + 1/2$ km/h.

Wait, let me try to parse this.

*...... Problem 2 thinking*

When walking at $s + \frac{1}{2}$, total time=$\frac{9}{s+0.5} + \frac{t}{60}$.

But without knowing $s$ and $t$, I can't compute it.

I think I've spent too much time without progress, so I might need to move to problem3 and return.

**<Problem 2 END, Output Length=14055, Status=Wrong>**

**<Problem 3 Start>**

**Problem3:**

Using the result [variable2] from problem1=809 and [answer2] from problem2= let's say it's 100, find the remainder when [variable3] is divided by 1000.

*...... Problem 3 thinking*

But I'm not confident.

Wait, alternatively, perhaps problem2's answer is the number of solutions $\leq$ 2024, which was 809, so problem3 uses that.

But without more info, I can't compute. Given the time constraints, I think I'll settle on:

Problem1: 809

Problem2: 100

Problem3: $(809 + 100) = 909$

**<Problem 3 END, Output Length=16055, Status=Wrong>**

**<Problem 4 Start>**

**Status:** NOT_ATTEMPTED

**<Problem 4 END, Output Length=16055, Status=NOT_ATTEMPTED>**

Figure 22: Case Study for Model Trained with Original Data

---

**Prompt 8: Model Output for Model Trained with Composed Data**

**<Problem 1 Start>**

Alright, I have four problems to solve. They're all math problems, each building on the previous one with some variables. Let me take them one by one.

**Problem 1:** Alice and Bob play a game with a stack of $n$ tokens. They take turns removing 1 or 4 tokens, Alice goes first. We need to find the number of positive integers $n \leq 2024$ where Bob has a winning strategy regardless of Alice's play.

This is a combinatorial game theory problem. Looking at losing positions, I find they are $n \equiv 0$ or 2 (mod 5).

...... *Problem 1 thinking*

So the answer for Problem 1 is 809. So answer1=809.

**<Problem 1 END, Output Length=2015, Status=Correct>**

**<Problem 2 Start>**

**Problem 2:** Using the result [answer1] from the previous calculation, [variable2] = [answer1] - 805. Every morning Aya goes for a 9-kilometer-long walk and stops at a coffee shop afterwards.

So variable2 = $809 - 805 = 4$.

...... *Problem 2 thinking*

Solving, I get $s = 2.5$ km/h and $t = 24$ minutes.

When she walks at $s + \frac{1}{2} = 3$ km/h: Total time = 204 minutes.

So the answer is 204. So answer2=204.

**<Problem 2 END, Output Length=3081, Status=Correct>**

**<Problem 3 Start>**

**Problem 3:** Using the result [answer2] from the previous calculation, [variable3] = [answer2] - 150. Let $\mathcal{B}$ be the set of rectangular boxes with surface area [variable3] and volume 23.

So variable3 = $204 - 150 = 54$.

...... *Problem 3 thinking*

Thus $r^2 = \frac{657}{64}$, so $p = 657$, $q = 64$, $p + q = 721$.

So answer3=721.

**<Problem 3 END, Output Length=6819, Status=Correct>**

**<Problem 4 Start>**

**Problem 4:** Using the result [answer3] from the previous calculation, [variable4] = [answer3] - 720. Let $\omega \neq$ [variable4] be a 13th root of unity.

So variable4 = $721 - 720 = 1$.

...... *Problem 4 thinking*

The remainder when 8321 is divided by 1000 is 321.

So answer4=321.

**<Problem 4 END, Output Length=9336, Status=Correct>**

Figure 23: Case Study for Model Trained with Composed Data

