# OpenReview forum: "R-Horizon: How Far Can Your Large Reasoning Model Really Go in Breadth and Depth?"
_ICLR.cc/2026/Conference — ICLR 2026 Poster_

### Official Review · Reviewer_91MQ · 2025-10-27

**Soundness:** 3
**Presentation:** 2
**Contribution:** 3
**Rating:** 6
**Confidence:** 3

**Summary:**

R-Horizon is motivated by existing LLM reasoning benchmarks' over-emphasis on _immediate, single-turn tasks_. It instead proposes a method to automatically generate _long-horizon, interdependent reasoning tasks_ via query composition, which enables it to curate the R-Horizon benchmark using existing (single-turn) datasets targetting math, coding, and agentic capabilities. The paper finds out that even the SOTA reasoning models suffer performance degradation as the number of composed queries (subproblems) increase. The paper then further suggests RLVR training over query-composed dataset, which demonstrates considerable improvement in both LRM's single- and multiple-problem performance. Extensive analysis and ablation studies provide important insights into the causes of performance degradation and why training with composed dataset works better.

**Strengths:**

- The paper is overall clearly motivated, well-written, and easy to follow.
- The lack of long-horizon, multi-stage/problem evaluation of LRMs is indeed a practical concern, and the paper's approach of using query composition to automatically combine single-turn queries within one interdependent problem is (an example of) an elegant solution.
- On top of the R-Horizon benchmark, the paper further conducts solid RLVR training on the composed dataset, which increases both single and multi-problem performance and further highlights the importance of equipping LRMs with multi-step compound reasoning skills.

**Weaknesses:**

- Even though the paper mentions that it constructs composed problems using both math, coding, and agentic datasets, most of the construction details and analysis are centering around the math criteria (see Q1 below). And the RL training is also solely done on math-related datasets.
- For the analysis on the RL training, it would be great to see similar "error type distribution" plot (as in Figure 5) for the model before & after the training. This helps gain further insights on _why & how the performance on multi-query problems_ increase over the training.
- While the writing is good overall, certain plots are hard to read due to ambiguous legends & captions. As an example, on Figure 8 the x-axis shows "Query 1/2/etc.", which I believe actually denotes the number of composed queries but can be misunderstood as query indices. Also certain plots have (unexplained) error bars/intervals while some do not.

**Questions:**

1. For the dataset construction for coding tasks, it's mentioned that _"unlike the sequential composition used for mathematical tasks, we apply a directly composed concatenation format ... without adding explicit dependencies"_. So (a) what is exactly this "directly composed concatenation format" (hard to tell simply from the given prompt in Appendix)? (b) isn't the point of R-Horizon benchmark to create interdependent, long-horizon reasoning tasks? I think the compromise made here undermines the overall validity (i.e. many of the analysis made in the paper, e.g. the "dependency reasoning error, applies to math only).
2. For the reward design, the distinction between the "last" and "all" rewards is a bit non-intuitive as I thought they should be very similar when the problems are inter-dependent. Is it even possible for a model to obtain positive "last" reward even when its "all" reward is 0 (the other direction is trivial)?
3. For **Figure 6**, how can the error positions be even larger than the output lengths (even query num is low)?

---

> ### Author Response · Authors · 2025-11-19
> **Response to Reviewer 91MQ (Part 1)**
>
> Thank you for your dedicated review of our R-HORIZON. In the following, we will carefully respond to your questions.
>
> ------
>
> ### **Response to Weakness 1 & Question 1: Different composition method on math, code and agentic task**
>
> **1. Diversity of Composition Forms in R-HORIZON**
>
> R-HORIZON incorporates multiple composition schemes, including Directly Compose, Sequential Compose, and Graphical Compose as illustrated in Figure 2(c).
>
> - Directly Compose: concatenation without explicit dependencies.
> - Sequential Compose: linear dependency chains.
> - Graphical Compose: graph-structured dependencies.
>
> For Math tasks, we use both Sequential Compose and Directly Compose (Appendix D.1 Ablation on Dependencies). For LiveCodeBench, we use Directly Compose. For Web Search task, we use graph-structured composition.
>
> **2. Why Code Tasks Do Not Use Dependency Construction**
>
> Our choice of composition strategies is based on the intrinsic properties of each task. In Math, both linear and graph-style dependencies can be constructed flexibly, and we found that linear dependencies already sufficiently reveal long-horizon difficulty, so we did not introduce unnecessary complexity. In Web Search tasks, dependencies arise naturally as graph structures (e.g., “Who is the architect of the Harry and Penelope Seidler House?” naturally decomposes into multiple linked nodes), making graphical composition appropriate.
>
> For LiveCodeBench, constructing dependencies is fundamentally challenging: (1) Extracting “key intermediate steps” from competitive programming problems is difficult. (2) Code evaluation relies on many test cases, and solving one problem does not produce a clean signal that can be passed to the next problem, making stable dependency construction impractical. (3) Real-world coding competitions often require solving *multiple independent tasks*, so Directly Compose still captures a realistic type of long-horizon reasoning for coding.
>
> Since R-HORIZON aims to simulate *realistic* long-horizon reasoning rather than force uniform dependencies across all tasks, we design composition strategies according to the characteristics of each domain.
>
> **3. Why the Analysis and RL Training Focus on Math**
>
> We focus heavily on Math because: (1) Math problems allow highly controlled and flexible composition, enabling clean variable-controlled analysis. (2) Math supports both linear and direct compositions, allowing us to clearly illustrate gaps between theoretical accuracy, linear-dependency accuracy, and independent-composition accuracy (Appendix D.1). (3) Constructing RL training data for Math is much cheaper, and there is substantial prior work (e.g., DAPO [1], Skywork-OR1 [2]) that provides practical guidance and baselines. In contrast, Code and Agent tasks are expensive to generate at scale, and introducing controlled dependencies is much more constrained. Thus, Math is the most suitable domain for detailed analysis and RL experiments.
>
> [1] DAPO: An Open-Source LLM Reinforcement Learning System at Scale
>
> [2] Skywork Open Reasoner 1 Technical Report

---

> ### Author Response · Authors · 2025-11-19
> **Response to Reviewer 91MQ (Part 2)**
>
> ### **Response to Weakness 2: Error Type Analysis Before & After the Training**
>
> Thank you for the suggestion. We agree that analyzing the **error type distribution** before and after training can provide deeper insights into why and how multi-query performance improves during training. Following your advice, we analyze three models—**R1-Qwen-7B**, **R1-Qwen-7B (n=1)**, and **R1-Qwen-7B (n=4)**—on Math500 with Query Num ∈ {2, 4, 8, 16} and AIME24 with Query Num ∈ {2, 3, 4, 5}. The aggregated error-type statistics for Math500 are shown in the table below:
>
> **Math500 Error Type Summary**
>
> | Model            | Correct | Problem Reasoning Error | Dependency Reasoning Error | Early Stop | Output Truncation |
> | ---------------- | ------- | ----------------------- | -------------------------- | ---------- | ----------------- |
> | R1-Qwen-7B       | 34.60%  | 50.45%                  | 7.70%                      | 7.10%      | 0.15%             |
> | R1-Qwen-7B (n=1) | 35.60%  | 52.20%                  | 6.95%                      | 4.90%      | 0.35%             |
> | R1-Qwen-7B (n=4) | 55.41%  | 41.24%                  | 2.70%                      | 0.30%      | 0.35%             |
>
> **AIME24** **Error Type Summary**
>
> | Model            | Correct | Problem Reasoning Error | Dependency Reasoning Error | Early Stop | Output Truncation |
> | ---------------- | ------- | ----------------------- | -------------------------- | ---------- | ----------------- |
> | R1-Qwen-7B       | 4.43%   | 86.98%                  | 2.94%                      | 5.57%      | 0.08%             |
> | R1-Qwen-7B (n=1) | 4.95%   | 84.87%                  | 2.87%                      | 6.95%      | 0.36%             |
> | R1-Qwen-7B (n=4) | 11.02%  | 87.11%                  | 1.15%                      | 0.55%      | 0.18%             |
>
> We observe that using **n = 4** composed data for RL training significantly improves accuracy and greatly reduces both **Dependency Reasoning Errors** and **Early Stop** cases when handling composed problems. The detailed error-type statistics by Query Num for Math500 are shown below:
>
>
> **Math500 Error Type Breakdown (by Query Num)**
>
> | Model            | Query Num | Correct | Problem Reasoning Error | Dependency Reasoning Error | Early Stop | Output Truncation |
> | ---------------- | --------- | ------- | ----------------------- | -------------------------- | ---------- | ----------------- |
> | R1-Qwen-7B       | 2         | 83.00%  | 14.00%                  | 2.40%                      | 0.60%      | 0.00%             |
> |                  | 4         | 43.60%  | 45.80%                  | 8.80%                      | 1.80%      | 0.00%             |
> |                  | 8         | 11.80%  | 67.60%                  | 12.20%                     | 7.80%      | 0.60%             |
> |                  | 16        | 0.00%   | 74.40%                  | 7.40%                      | 18.20%     | 0.00%             |
> | R1-Qwen-7B (n=1) | 2         | 88.40%  | 10.40%                  | 0.40%                      | 0.40%      | 0.40%             |
> |                  | 4         | 46.40%  | 43.80%                  | 7.40%                      | 2.00%      | 0.40%             |
> |                  | 8         | 7.60%   | 76.20%                  | 11.40%                     | 4.80%      | 0.00%             |
> |                  | 16        | 0.00%   | 78.40%                  | 8.60%                      | 12.40%     | 0.60%             |
> | R1-Qwen-7B (n=4) | 2         | 90.20%  | 9.20%                   | 0.60%                      | 0.00%      | 0.00%             |
> |                  | 4         | 76.40%  | 23.00%                  | 0.60%                      | 0.00%      | 0.00%             |
> |                  | 8         | 53.40%  | 43.80%                  | 2.20%                      | 0.40%      | 0.20%             |
> |                  | 16        | 1.41%   | 89.16%                  | 7.43%                      | 0.80%      | 1.20%             |
>
> ---
>
> ### **Response to Weakness 3: Ambiguous Legends & Captions in Figures**
>
> Thank you for pointing this out. We reduced many captions to save space, which unfortunately made some plots harder to interpret. We have now fixed these issues:
>
> - In Figure 8, we corrected the x-axis label from *“Query 2”* to *“2”*.
> - In Figure 7, we clarified the previously unexplained error bars and added the corresponding explanation in the caption.
>
> These updates improve the readability and interpretability of the figures. You can find these updates in our submitted revision.

---

> ### Author Response · Authors · 2025-11-19
> **Response to Reviewer 91MQ (Part 3)**
>
> ### **Response to Question 2: Reward Design for all and last problem**
>
> Yes, it is indeed possible for a model to obtain a positive **“last”** reward even when its **“all”** reward is 0. During evaluation, we observed an anomalous behavior: the model can sometimes answer later questions correctly even after failing earlier ones. This indicates that the model occasionally produces correct outputs for problems that should be unsolvable due to broken dependencies.
>
> We analyze this phenomenon in detail in **Appendix D.2**, and we report the discrepancy between **Acc_last** and **Acc_all** in the following table (for R1-Qwen-7B and R1-Qwen-32B on Math500):
>
> *Acc_last vs. Acc_all Comparison*
>
> | **query_num** | **7B Acc_all** | **7B Acc_last** | **7B anomaly examples** | **32B Acc_all** | **32B Acc_last** | **32B anomaly examples** |
> | ------------- | -------------- | --------------- | ----------------------- | --------------- | ---------------- | ------------------------ |
> | 2             | 0.8300         | 0.8360          | 3                       | 0.9300          | 0.9360           | 3                        |
> | 4             | 0.4360         | 0.5060          | 59                      | 0.8300          | 0.8720           | 22                       |
> | 8             | 0.1180         | 0.1700          | 123                     | 0.6540          | 0.7780           | 76                       |
> | 12            | 0.0040         | 0.0120          | 144                     | 0.5540          | 0.7700           | 129                      |
> | 16            | 0.0000         | 0.0060          | 112                     | 0.3760          | 0.7140           | 210                      |
> | 20            | 0.0000         | 0.0020          | 97                      | 0.2540          | 0.6580           | 283                      |
>
> As shown in the table, the number of anomalous cases increases substantially as the number of composed problems grows, for both the 7B and 32B models. For the 32B model, the gap between Acc_all and Acc_last reaches 40% at Query 20. We believe this phenomenon is primarily caused by data contamination [1]. This finding motivated us to design both Reward_all and Reward_last to minimize such abnormal reward signals. As reported in Table 1 in paper, training with the stricter Reward_all leads to better overall performance.
>
> [1] Reasoning or Memorization? Unreliable Results of Reinforcement Learning Due to Data Contamination
>
> ---
>
> ### **Response to Question 3: Error Positions Larger than the Output Length**
>
> In Figure 6 (Query Num = 1), incorrect problems are generally longer than correct ones, but the number of incorrect answers on Math500 is very small compared to the number of correct answers (32 wrong vs 468 correct on R1-Qwen-7B). Since the average error position is higher than the average length of correct problems, this leads to the error position appearing above the average token length.

---

> > ### Comment · Reviewer_91MQ · 2025-11-22
> > **Thanks for the clarification**
> >
> > Dear authors,
> >
> > Thanks for giving a clear and concise point-by-point rebuttal to the weakness & questions I've raised. I particularly like how the concrete breakdowns in error type analysis & acc_last/acc_all discrepancies let me gain some insights better. In general I think most of my questions are answered.
> >
> > As a result, I'm happy to increase my score and the soundness assessment of this paper.
> >
> > I still think that it might be possible to add dependencies into coding questions? After all LiveCodeBench contains many Leetcode/Codeforces-like questions -- so I imagine similar strategies in math, such as replacing the value of one input variable (if it exists) by the solution of a previous question -- might work here? I think there's no pressure to include these during the rebuttal window, but would genuinely love to see more interdependent & challenging reasoning tasks to be curated to assess the limit of LRMs.
> >
> > -- Reviewer 91MQ

---

> > > ### Author Response · Authors · 2025-11-22
> > > **Thanks for the response and valuable suggestions**
> > >
> > > Dear Reviewer 91MQ,
> > >
> > > We are very glad that you found our rebuttal helpful, and we sincerely appreciate your decision to increase the score and the soundness assessment of our paper.
> > >
> > > Regarding your excellent suggestion on introducing dependencies into coding tasks: we indeed attempted to construct compositional coding problems, but encountered several practical challenges. As you mentioned, LiveCodeBench contains many Leetcode/Codeforces-style questions, and in principle it is possible to create dependencies by replacing certain input variables with the solution to a previous question. Although identifying such variables is harder than in math tasks, it remains feasible.
> > >
> > > **The primary challenge lies in constructing reliable answer signals for compositional Code problems.**
> > > Unlike math tasks—where the model produces a single numeric answer that can be directly passed to the next question—coding tasks require generating full code snippets whose correctness can only be verified by executing them in an external sandbox with many test cases. This means that after each sub-problem in a compositional chain, the model would need external evaluation to determine correctness (or obtain answer signals), and such a workflow is hard for current models to internalize without additional specialized training. Moreover, the sandbox provides only a binary correct/incorrect signal, lacking the richer intermediate answer signals that naturally support composition in math.
> > >
> > > Despite these challenges, we agree that constructing compositional coding tasks is feasible with better design and optimization. We share your enthusiasm and genuinely would also love to see more interdependent and challenging reasoning tasks developed to assess the limits of LRMs. We hope to improve this aspect in future work.
> > >
> > > Thank you again for your constructive feedback, your valuable suggestions, and your recognition of our work.

---

### Official Review · Reviewer_sdPJ · 2025-10-27

**Soundness:** 3
**Presentation:** 3
**Contribution:** 3
**Rating:** 6
**Confidence:** 3

**Summary:**

This paper introduces R-HORIZON, a benchmark and training framework designed to evaluate and enhance long-horizon reasoning in Large Reasoning Models (LRMs).
Existing benchmarks mainly assess single-step reasoning, where each question is independent. R-HORIZON addresses this gap by composing interdependent multi-step queries from existing datasets (e.g., MATH500, AIME24/25, LiveCodeBench, WebShaper).
The authors evaluate 25 state-of-the-art reasoning models (e.g., DeepSeek-R1, Qwen-3-235B, Gemini-2.5, Claude-Sonnet-4) and show that all suffer severe performance degradation as reasoning horizons increase.
Furthermore, they train models using R-HORIZON-based reinforcement learning (RLVR), demonstrating substantial gains on both composed and standard reasoning tasks (+17.4 on long-horizon tasks and +7.5 on AIME24).

Overall, the paper provides a new perspective on reasoning length, reflection behavior, and token budget allocation, revealing fundamental limits in current LRMs and proposing a practical solution for improvement.

**Strengths:**

Novel and impactful evaluation paradigm.
R-HORIZON systematically measures reasoning depth and breadth, exposing weaknesses that single-horizon benchmarks cannot reveal.

Comprehensive empirical analysis.
The evaluation across 25 strong LRMs and multiple domains (math, code, agentic tasks) convincingly demonstrates consistent long-horizon degradation patterns.

Actionable training insights.
The reinforcement learning experiments with R-HORIZON data provide a scalable and cost-effective way to enhance long-horizon reasoning, with clear empirical gains and behavioral analyses (reflection, budget allocation).

**Weaknesses:**

Multi-step reasoning is indeed necessary; however, in this paper, simply combining several basic math problems offers limited reflection of contextual consistency in true multi-step reasoning. Tasks such as “1 + 2 = ?” followed by “previous answer + 4 = ?” can essentially be summarized into a single query like “1 + 2 + 4 = ?”, which weakens the intended long-horizon reasoning challenge.

Moreover, maintaining contextual consistency across multi-step reasoning is highly dependent on prompt design, yet the paper does not clearly explain how prompts are constructed or formatted for solving these composed problems.

The evaluation on closed-source models is insufficient, as only o4-mini was included.

**Questions:**

None

---

> ### Author Response · Authors · 2025-11-19
> **Response to Reviewer sdPJ (Part 1)**
>
> Thank you for your dedicated review of our R-HORIZON. In the following, we will carefully respond to your questions.
>
> ------
>
> ### **Response to Weakness on Problem Composition**
>
> Our goal with R-HORIZON is not only to create a challenging benchmark, but to *isolate and understand* the gap between single-step reasoning and long-horizon reasoning. To do this, we deliberately use simple and transparent dependencies when composing problems. In the math domain, these dependencies are basic additions or subtractions, allowing us to avoid increasing inherent task difficulty and instead focus on the effect of *horizon length* and *dependency chains*.
>
> Because each subproblem remains simple, we can analytically compute the theoretical accuracy of multi-step compositions from single-step accuracy. This makes it possible to compare *actual accuracy vs. theoretical accuracy* (Figure 1) and quantify how much performance degrades purely due to long-horizon reasoning.
>
> Importantly, in long-horizon settings, tasks like “1+2=?” followed by “previous answer + 4=?” cannot be reduced to a single query such as “1+2+4=?”. Even minimal dependencies force the model to:
>
> 1. correctly solve each step sequentially, and
> 2. reliably propagate intermediate results across the chain.
>
> Our experiments show that these factors substantially increase difficulty. In Appendix D.1 (Ablation on Dependencies), we compare *independent composition* with *composition involving dependencies*. Below is the table from Figure 12 (R1-Qwen-7B on Math500):
>
> | query_num                 | 2      | 4      | 6      | 8      | 10     |
> | ------------------------- | ------ | ------ | ------ | ------ | ------ |
> | Independent Compose       | 0.8700 | 0.8240 | 0.7420 | 0.5940 | 0.5440 |
> | Compose with Dependencies | 0.832  | 0.436  | 0.208  | 0.118  | 0.026  |
> | Expected Accuracy         | 0.9020 | 0.8620 | 0.7780 | 0.7160 | 0.6800 |
>
> We find that the accuracy of both problem composition methods falls below the expected accuracy, and the accuracy of multiple sequentially dependent problems is significantly lower than that of multiple independent problems，demonstrating that even very simple dependencies create substantial challenges that cannot be reduced to single-query equivalents.
>
> These results highlight that simple mathematical dependencies are sufficient to capture essential difficulties of long-horizon reasoning while remaining analyzable, controlled, and interpretable.
>
> ---
>
> ### **Response to Weakness on Prompt Design**
>
> We agree that prompt design plays a crucial role in simulating multi-step reasoning. In Section 3.1, we formally describe how composed problems are constructed, and in Appendix E.1 we provide concrete prompt templates for the math, code, and web search tasks.
>
> To address this concern more explicitly, we have updated Section 4.1 in the revised manuscript (highlighted in blue) to point readers to the full prompt details. These additions clarify how prompts are structured and ensure full transparency of our multi-step reasoning setup.

---

> ### Author Response · Authors · 2025-11-19
> **Response to Reviewer sdPJ (Part 2)**
>
> ### **Response to Weakness on Insufficient Closed-Source Models**
>
> We included o4-mini, Gemini-2.5-Flash-Thinking, and Claude-Sonnet-4 in Figure 3. We understand the reviewer’s request to see results on the most advanced models. In addition to those closed-source models, we have now added evaluations for Gemini-2.5-Pro. The closed-source model results are listed below.
>
> |                           | Math500 |         |         |         |          | AIME24 |         |         |         |         | AIME25 |         |         |         |         |
> | ------------------------- | ------- | ------- | ------- | ------- | -------- | ------ | ------- | ------- | ------- | ------- | ------ | ------- | ------- | ------- | ------- |
> |                           | Single  | Multi-2 | Multi-4 | Multi-8 | Multi-16 | Single | Multi-2 | Multi-3 | Multi-4 | Multi-5 | Single | Multi-2 | Multi-3 | Multi-4 | Multi-5 |
> | Gemini-2.5 Pro            | 97.6%   | 94.0%   | 84.0%   | 70.4%   | 64.6%    | 82.0%  | 59.1%   | 43.8%   | 53.1%   | 40.7%   | 73.7%  | 50.2%   | 37.8%   | 42.2%   | 23.8%   |
> | Gemini-2.5-Flash-Thinking | 98.4%   | 92.6%   | 80.2%   | 67.8%   | 45.6%    | 74.3%  | 46.1%   | 20.1%   | 18.4%   | 12.0%   | 53.7%  | 22.2%   | 9.0%    | 9.9%    | 2.4%    |
> | o4-mini                   | 98.4%   | 96.8%   | 90.8%   | 85.0%   | 73.4%    | 82.3%  | 63.6%   | 40.4%   | 44.1%   | 30.5%   | 79.7%  | 50.2%   | 37.3%   | 48.8%   | 16.5%   |
> | anthropic.claude-sonnet-4 | 86.6%   | 89.0%   | 82.2%   | 69.4%   | 50.8%    | 46.3%  | 34.1%   | 32.8%   | 31.0%   | 9.0%    | 70.5%  | 21.8%   | 21.7%   | 22.3%   | 2.7%    |
>
>
>
> |                           | AMC23  |         |         |         |         | LiveCodeBench |         |         |         |         | WebShaper |         |         |         |         |
> | ------------------------- | ------ | ------- | ------- | ------- | ------- | ------------- | ------- | ------- | ------- | ------- | --------- | ------- | ------- | ------- | ------- |
> |                           | Single | Multi-2 | Multi-4 | Multi-6 | Multi-8 | Single        | Multi-2 | Multi-3 | Multi-4 | Multi-5 | Single    | Multi-2 | Multi-3 | Multi-4 | Multi-5 |
> | Gemini-2.5 Pro            | 97.5%  | 85.9%   | 77.5%   | 79.4%   | 70.6%   | 68.8%         | 33.3%   | 15.4%   | 6.5%    | 3.6%    | 76.2%     | 77.8%   | 70.7%   | 65.3%   | 64.0%   |
> | Gemini-2.5-Flash-Thinking | 95.3%  | 59.4%   | 55.6%   | 51.9%   | 29.7%   | 45.5%         | 18.6%   | 6.8%    | 1.4%    | 1.1%    | 45.3%     | 56.5%   | 55.3%   | 48.7%   | 47.3%   |
> | o4-mini                   | 100.0% | 91.6%   | 90.6%   | 84.4%   | 80.9%   | 76.3%         | 39.8%   | 20.1%   | 13.3%   | 10.0%   | 83.7%     | 82.6%   | 84.4%   | 76.3%   | 81.4%   |
> | anthropic.claude-sonnet-4 | 81.6%  | 60.0%   | 42.2%   | 30.6%   | 17.2%   | 50.9%         | 11.8%   | 4.3%    | 2.5%    | 1.4%    | 48.0%     | 43.3%   | 35.3%   | 35.3%   | 40.7%   |
>
> These results show that even the most advanced closed-source models still struggle significantly with long-horizon reasoning. We will update Figure 3 in the paper to include the newly added results for Gemini-2.5-Pro.

---

> ### Author Response · Authors · 2025-11-27
> **Follow up Response to Reviewer sdPJ**
>
> Dear Reviewer sdPJ,
>
> We would like to express our sincere gratitude for your thorough review of our manuscript and for your valuable insights. We have carefully considered your feedback and have made significant revisions to address your concerns.
>
> In particular, we have:
>
> - Clarified the rationale behind our problem composition strategy in R-HORIZON. By deliberately using simple dependencies, we isolate long-horizon reasoning effects without confounding task difficulty. Our experiments (Appendix D.1) demonstrate that even minimal sequential dependencies substantially increase model error rates compared to independent compositions or single-step queries.
> - Strengthened evaluation on closed-source models by adding results for Gemini-2.5-Pro alongside o4-mini, Gemini-2.5-Flash-Thinking, and Claude-Sonnet-4 as shown in Figure 3 and detailed tables above; these new results confirm that even state-of-the-art proprietary LRMs face significant challenges with long-horizon reasoning.
>
> We kindly ask if you could take a moment to review our updated manuscript. If there is anything further that you would like us to clarify or any additional feedback you wish to provide, please let us know. We are more than willing to provide any additional information or make further revisions as needed.
>
> Thank you once again for your time and thoughtful feedback. Your insights have been instrumental in improving our paper.
>
> Sincerely,
>
> Authors of Submission 6849

---

### Official Review · Reviewer_JsjS · 2025-10-30

**Soundness:** 3
**Presentation:** 3
**Contribution:** 3
**Rating:** 6
**Confidence:** 5

**Summary:**

The paper proposes R-HORIZON, a benchmark and data construction framework for evaluating and improving long-horizon reasoning in large reasoning models (LRMs). Using Expanded Problem Composition (EPC), it links independent problems into multi-step dependency chains. Evaluating 25 LRMs across math, code, and agentic tasks, the authors find significant performance degradation as reasoning length increases. They further integrate R-HORIZON data into RL from verifiable rewards (RLVR) using Group Relative Policy Optimization (GRPO), showing that such training enhances models’ sustained reasoning ability.

**Strengths:**

**Strengths:**

1. The paper is well-written, clearly structured, and easy to follow.
2. It introduces meaningful long-horizon benchmarks and corresponding datasets that are valuable for advancing research on large reasoning models (LRMs).
3. The extended benchmarks effectively reveal the limitations of current LRMs in maintaining long-horizon reasoning.
4. The authors conduct reinforcement learning (RL) experiments on the constructed datasets, demonstrating notable improvements in sustained reasoning performance.
5. The paper provides comprehensive analyses and insights into the proposed data construction and evaluation methods.

**Weaknesses:**

**Weaknesses:**
1. While the proposed methods enable verifiable long-horizon reasoning evaluation, the constructed benchmarks appear somewhat artificial and may not fully reflect realistic reasoning scenarios.
2. Similar efforts, such as *GSM-Infinity* [1], have explored benchmark construction for increasing reasoning complexity. The paper would benefit from a clearer discussion of how its approach differs from these prior works.

[1] GSM-Infinite: *How Do Your LLMs Behave over Infinitely Increasing Context Length and Reasoning Complexity?* (ICML 2025) https://arxiv.org/abs/2502.05252

**Questions:**

**Questions for the Authors:**
1. In **Figure 11**, the entropy of RL training decreases as *n* increases (e.g., *n=4* vs. *n=1*). Could the authors provide an explanation or hypothesis for why longer reasoning chains lead to lower entropy?
2. Is it possible that the poor performance of models on **R-HORIZON** arises because they have not been exposed to similar long-horizon or compositional data during pre-training or intermediate training? Have you attempted any supervised fine-tuning (SFT) or cold-start experiments on such data?
3. Could the improvement in original tasks after RL with R-HORIZON stem from the fact that the constructed datasets yield more balanced reward signals (i.e., less likely to produce all-correct or all-wrong rollouts), thereby improving rollout efficiency?
4. The paper reports an increase in reasoning efficiency (shorter thinking length) after RL. Could this be because higher *n* naturally leads to longer responses, which increases the truncation ratio during RL training, thus implicitly encouraging more concise reasoning?

---

> ### Author Response · Authors · 2025-11-19
> **Response to Reviewer JsjS (Part 1)**
>
> Thank you for your dedicated review of our R-HORIZON. In the following, we will carefully respond to your questions.
>
> ------
>
> ### **Response to Weakness 1: The Gap Between R-HORIZON  and Realistic Reasoning Scenarios**
>
> Our primary motivation for R-HORIZON is to **close the gap between existing single-step benchmarks and realistic long-horizon reasoning**. Real-world reasoning typically involves **sequences of interdependent steps**, yet current datasets evaluate only isolated problems and therefore cannot reflect a model’s ability to maintain consistency or allocate reasoning budget over long trajectories.
>
> We acknowledge that purely synthetic mathematical compositions may still differ from real-world tasks. To address this, we intentionally incorporate **more realistic tasks such as web search and agentic environments**, where long-horizon dependencies arise naturally. This allows R-HORIZON to better approximate practical multi-step reasoning while remaining **verifiable, scalable, and controlled**—properties real-world datasets often lack due to their high collection cost and difficulty of step-by-step validation.
>
> In summary, while no benchmark can fully replicate real-world conditions, R-HORIZON significantly narrows the gap compared to existing single-horizon evaluations and provides the first systematic, verifiable framework for studying long-horizon reasoning.
>
> ---
>
> ### **Response to Weakness 2: Distinction from GSM-Infinite**
>
> Thank you for the suggestion. Although GSM-Infinite [1] shares a similar idea of compositional construction, it differs from R-HORIZON in both methodology and target scenarios.
>
> - In terms of methodology, both GSM-Infinite and R-HORIZON abstract reasoning tasks into computation graphs. However, GSM-Infinite focuses on composing *internal sub-steps within a single problem*, whereas R-HORIZON composes *across different problems*. While GSM-Infinite can precisely control the number of steps, the sub-steps it creates are intentionally very simple. When many such homogeneous steps are combined, the resulting tasks deviate substantially from realistic reasoning (e.g., GSM-Infinite can only combine +, –, ×, ÷, whereas real mathematical reasoning is far more complex). R-HORIZON preserves the full structure of original problems and evaluates whether models can solve multiple problems sequentially.
>
> - GSM-Infinite mainly targets **long-context inputs**, while R-HORIZON focuses on **long outputs**. GSM-Infinite introduces noise by adding nodes and edges to create long input contexts, showing that reasoning degrades with longer inputs. In contrast, R-HORIZON evaluates performance under long reasoning trajectories (Section 5.1 Effective Reasoning Length), which better reflects real scenarios where inputs are short but Chain-of-Thought can be very long.
>
> We have added a concise comparison to GSM-Infinite in the Related Work section.
>
> [1] GSM-Infinite: How Do Your LLMs Behave over Infinitely Increasing Context Length and Reasoning Complexity? (ICML 2025)

---

> ### Author Response · Authors · 2025-11-19
> **Response to Reviewer JsjS (Part 2)**
>
> ### **Response to Question 1: Entropy Decrease as the number of composed problem increases**
>
> We hypothesize that as the number of problems increases, each problem receives a smaller thinking budget, which reduces the model’s exploration space during reasoning. To verify this, we measure the model’s reflection frequency and the proportion of reflection-related tokens. These reflection-related tokens are typically **high-entropy tokens** [1], so their proportion provides an approximate indicator of the model’s exploration during thinking.
>
> We analyze the reflection behavior and the proportion of high-entropy reflection tokens (e.g., *wait, but, however, let…*) for R1-Qwen-7B trained with **n = 1** and **n = 4**. The results are shown below:
>
> **Math500 Results**
>
> | **# Composed Query** | **n=1 Reflection Freq** | **n=4 Reflection Freq** | **n=1 Reflection Token %** | **n=4 Reflection Token %** | **n=1 Accuracy** | **n=4 Accuracy** |
> | -------------------- | ----------------------- | ----------------------- | -------------------------- | -------------------------- | ---------------- | ---------------- |
> | 2                    | 38,907                  | 11,272                  | 2.0545                     | 1.3119                     | 0.88             | 0.922            |
> | 4                    | 86,633                  | 21,595                  | 2.0650                     | 1.3864                     | 0.524            | 0.732            |
> | 8                    | 144,394                 | 37,151                  | 2.0649                     | 1.3333                     | 0.084            | 0.214            |
> | 12                   | 162,143                 | 59,062                  | 2.0721                     | 1.2407                     | 0.00             | 0.012            |
>
> As shown in the table, when the number of queries increases, the model trained with n = 4 exhibits significantly lower reflection frequency and a lower proportion of high-entropy reflection tokens compared with n = 1. Even under similar accuracy (e.g., Math500 2 composed query), the reflection frequency of the n=1 model is 4× higher, and its proportion of high-entropy reflection tokens is 2× higher than that of the n=4 model. This supports our hypothesis that, under composition training, the model’s exploration capacity diminishes in later training stages due to the limited per-problem token budget.
>
> [1] Beyond the 80/20 Rule: High-Entropy Minority Tokens Drive Effective Reinforcement Learning for LLM Reasoning
>
> ---
>
> ### **Response to Question 2: Impact of Supervised Fine-tuning (SFT) on R-HORIZON Performance**
>
> Yes, we agree that the poor performance of models on R-HORIZON is closely related to the fact that they have never been exposed to similar long-horizon or compositional data during pre-training or intermediate training. Motivated by this insight, we conducted an additional experiment using **Rejection Fine-Tuning (RFT)** on **R1-distill-Qwen-7B**. The procedure is as follows:
>
> 1. We constructed **120k** compositional samples by applying the **n = 4** composition setting to the **OpenR1-Math-94k** [1] dataset.
> 2. We distilled the n=4 model (R1-Qwen-7B trained with n=4 data in Section 4.3) on this composed dataset using rejection sampling.
> 3. We then performed **supervised fine-tuning (SFT)** on R1-distill-Qwen-7B using the filtered correct trajectories and evaluated the resulting model.
>
> |                           | Math500 |         |         |         |          | AIME24 |         |         |         |         | AIME25 |         |         |         |         | AMC    |         |         |         |         |
> | ------------------------- | ------- | ------- | ------- | ------- | -------- | ------ | ------- | ------- | ------- | ------- | ------ | ------- | ------- | ------- | ------- | ------ | ------- | ------- | ------- | ------- |
> |                           | Single  | Multi-2 | Multi-4 | Multi-8 | Multi-16 | Single | Multi-2 | Multi-3 | Multi-4 | Multi-5 | Single | Multi-2 | Multi-3 | Multi-4 | Multi-5 | Single | Multi-2 | Multi-4 | Multi-6 | Multi-8 |
> | R1dist-Qwen-7B            | 93.6%   | 83.2%   | 43.6%   | 11.8%   | 0.0%     | 48.3%  | 16.4%   | 1.1%    | 0.1%    | 0.0%    | 33.3%  | 3.5%    | 0.0%    | 0.0%    | 0.0%    | 89.9%  | 48.8%   | 6.2%    | 0.3%    | 0.0%    |
> | R1dist-Qwen-7B-n4_distill | 95.4%   | 91.8%   | 77.8%   | 53.0%   | 1.6%     | 53.1%  | 32.4%   | 5.6%    | 4.4%    | 0.4%    | 33.9%  | 9.2%    | 1.5%    | 2.6%    | 0.0%    | 91.9%  | 77.5%   | 55.0%   | 29.1%   | 2.8%    |
>
> We observe that after performing SFT on the distilled data, the model’s performance improves in both the single-problem and compositional-problem settings, with the gains being notably larger for compositional problems.
>
> [1] OpenR1-Math-94k: https://huggingface.co/datasets/llamafactory/OpenR1-Math-94k

---

> ### Author Response · Authors · 2025-11-19
> **Response to Reviewer JsjS (Part 3)**
>
> ### **Response to Question 3: Role of Balanced Reward Signals in RL Improvements**
>
> We compute the proportion of *Solve None*, *Solve All*, and *Effective* samples in each training batch. As shown in the table below, models trained with compositional data using **n = 2** and **n = 4** obtain, on average, **20% more effective samples** compared with **n = 1**. This indicates that the constructed datasets indeed yield more balanced reward signals. By combining multiple problems, the model receives a larger amount of effective training data, thereby improving rollout efficiency. We have added this part to the Analysis on Rollout Efficiency in our revision.
>
> | Step    | Model   | Solve None (%) | Solve All (%) | Effective (%) |
> | ------- | ------- | -------------- | ------------- | ------------- |
> | 100     | n=1     | 3.9            | 16.8          | 79.3          |
> |         | n=2     | 6.6            | 2.7           | 90.6          |
> |         | n=4     | 27.0           | 0.0           | 73.0          |
> | 200     | n=1     | 6.2            | 23.4          | 70.3          |
> |         | n=2     | 0.4            | 6.6           | 93.0          |
> |         | n=4     | 7.0            | 0.4           | 92.6          |
> | 300     | n=1     | 4.7            | 28.9          | 66.4          |
> |         | n=2     | 2.0            | 8.6           | 89.5          |
> |         | n=4     | 7.0            | 2.0           | 91.0          |
> | 400     | n=1     | 2.0            | 34.0          | 64.1          |
> |         | n=2     | 2.7            | 17.2          | 80.1          |
> |         | n=4     | 1.6            | 3.5           | 94.9          |
> | 500     | n=1     | 2.3            | 32.8          | 64.8          |
> |         | n=2     | 1.2            | 18.8          | 80.1          |
> |         | n=4     | 2.7            | 7.8           | 89.5          |
> | 600     | n=1     | 4.7            | 35.2          | 60.2          |
> |         | n=2     | 1.2            | 28.5          | 70.3          |
> |         | n=4     | 2.7            | 9.4           | 87.9          |
> | **Avg** | **n=1** | **4.0**        | **28.5**      | **67.5**      |
> |         | **n=2** | **2.3**        | **13.7**      | **83.9**      |
> |         | **n=4** | **8.0**        | **3.8**       | **88.2**      |
>
>
> ---
>
> ### **Response to Question 4: Reasoning Efficiency and Truncation Ratio Effects**
>
> Yes. We observe that when using compositional data, the **response length decreases rapidly in early training** (Figure 11). This is essentially because the maximum output length constrains the model’s reasoning capacity, similar to the 8k stage in the multi-stage training of Skywork-OR1 [1]. During training, the model learns to **allocate its token budget more efficiently** and to solve the problem within a limited budget.
>
> [1] Skywork Open Reasoner 1 Technical Report

---

> ### Author Response · Authors · 2025-11-27
> **Follow up Response to Reviewer JsjS**
>
> Dear Reviewer JsjS,
>
> We would like to express our sincere gratitude for your thorough review of our manuscript and for your valuable insights. We have carefully considered your feedback and have made significant revisions to address your concerns.
>
> Specifically, we have:
>
> - Clarified the motivation behind R-HORIZON, emphasizing its aim to bridge the gap between single-step benchmarks and realistic long-horizon reasoning scenarios. We further distinguished R-HORIZON from GSM-Infinite in both methodology (cross-problem composition vs. intra-problem substep composition) and evaluation focus (long output trajectories vs. long input contexts).
> - Added additional experiments addressing your questions:
>   - Provided a detailed analysis of entropy reduction during RL training with increasing number of problem composition, including reflection token statistics.
>   - Included supervised fine-tuning results showing that exposure to compositional data substantially improves model performance on multi-step tasks.
>   - Quantified how balanced reward signals in constructed datasets enhance rollout efficiency during RL.
>
> We kindly ask if you could take a moment to review our updated manuscript. If there is anything further that you would like us to clarify or any additional feedback you wish to provide, please let us know. We are more than willing to provide any additional information or make further revisions as needed.
>
> Thank you once again for your time and thoughtful feedback. Your insights have been instrumental in improving our paper.
>
> Sincerely,
>
> Authors of Submission 6849

---

### Official Review · Reviewer_hTcg · 2025-10-31

**Soundness:** 3
**Presentation:** 3
**Contribution:** 3
**Rating:** 6
**Confidence:** 3

**Summary:**

The authors propose a new dataset for long-horizon reasoning and then also train a model on their dataset with Reinforcement Learning (RL). The new dataset is based on six other datasets targeting math (MATH500, AMC23, AIME24 and AIME25), code generation (LiveCodeBench) and agentic tool usage (WebShaper). Long-horizon problems are generated by combining several independent problems into a single one, where the problems have to be solved (usually) sequentially since later problems rely on the results of earlier problems, thereby transforming them into a long-horizon task. 25 language models are then evaluated on the developed dataset and their results are analyzed. In addition the author also generate a training dataset based on the Skywork-RL dataset and use it to fine-tune an existing model with custom reward functions, which then is also evaluated.

**Strengths:**

* with the increase in context lengths of modern language models, long-horizon reasoning is a relevant topic and the dataset seems to be good contribution
* the writing is mostly clear
* good evaluation in general and large number of RLMs tested
* good ablation study
* detailed appendix with good overview over additional results as well as the evaluation setup

**Weaknesses:**

* I found the motivation a bit lacking: the new dataset is somewhat artificial in its dependencies - a proper motivational example would be beneficiary in my opinion
* not a weakness per se, but I found the third chapter a bit too high-level without much details
* writing primarily in the evaluation is at times a bit imprecise or not entirely accurate:
  * 4.1:
    * AMC23 is not discussed
    * for MATH500 it seems that there is also data for 20 composed query numbers
  * 4.2, first paragraph: "R1-Qwen-7B drops from 93.6% (n = 1) to 0% (n = 16), which is 34.1% more than the 32B model" - reference is unclear (probably MATH500); Where does the 34.1% come from?
  * 5.1, Effective Reasoning Length of LRMs:
    * "gap between the actual accuracy and theoretical accuracy of models becomes increasingly larger" - not necessarily true on AIME24
    * "7B model’s error range is (4-6k tokens) while the 32B model’s error range is (8-10k tokens)" - context is missing, probably for MATH500
  * Reflection is not defined or at least explained, which also makes it unclear why it matters.
  * Sometimes "expected accuracy" and sometimes "theorectical accuracy" is used? I assume they are the same thing, so I suggest to use one term consistently.
  * Appendix D.3: "Figure 14 (b) and (c) show that all models fail to allocate thinking budget reasonably according to problem difficulty" - seems to be somewhat not true for R1-Qwen-7B

minor issues:
* introduction: no reference for CoT
* Figure 1: Theoretical Accuracy is not introduced at this point. (only on page 4)
* 2.2:
  * "Su et al. (2025); Yang et al. (2025b); Wu et al. (2025b) investigates..." - It should be "investigate", since it is a plural.
* Algorithm 1: "and Create" - lowercase create?
* 4.1: I believe, there is no whitespace before a footnote.
* 4.3, Impact of Number of Composed Queries and Different Reward Schemes: n=1 is not really a composed problem
* Figure 5: not very readable in terms of fontsize as well as chosen color for example for early stop and output truncation - I had to zoom in to 400% to find an instance for output truncation
  * similar issue for Figure 6, especially the theoretical accuracy as well as Figures 7 to 9
* Figure 6: missing whitespace before the brackets in the subfigure titles (also for Figure 9)
* sometimes it is "Math500" and sometimes it is "MATH500" in the evaluation figures
* Figure 8: x axis label might be misleading, maybe something like "2 Queries" or just "2" and so on would be better?
* Appendix F.1: "policy loss 5 with" - should probably be something like "policy loss (see Eq. 5) with"
* Appendix H: details on model, which generated the responses are missing, where the questions are coming from, etc.
* references:
  * consider the proper capitalization of the titles, at least for proper names and abbreviations to improve readability
  * place of publication of arXiv references can only surmised from the URL
  * consider adding the access date for web references
  * [Guo et al. 2025] - cited differently than the other arXiv references
  * [Luo et al. 2025a/b] - URL is not a proper link
  * [MAA 2024/2025] - consider using something like the xurl package to fix the formatting
  * [Muennighoff et al. 2025] - cited differently than the other arXiv references - maybe this could be the blueprint for the others

**Questions:**

* Figure 2: What are the circles with a red outline?
* 4.2, last paragraph: Why is AIME25 more challenging than AIME24?
* Figure 6: How can the error position be above the token length? Because wrong answers tend to be longer for smaller query numbers?
* How much can the six benchmarks (MATH500, AMC23, AIME24/25, LiveCodeBench and WebShaper) be considered out of distribution compared to the training dataset based on Skywork-RL?

---

> ### Author Response · Authors · 2025-11-19
> **Response to Reviewer hTcg (Part 1)**
>
> Thank you for your dedicated review of our R-HORIZON. In the following, we will carefully respond to your questions.
>
> ------
>
> ### **Response to Weakness on Motivation**
>
> Our motivation is to address a fundamental gap in current reasoning benchmarks: **they evaluate only single, isolated problems**, whereas realistic reasoning requires **solving a sequence of interdependent tasks** over a long horizon.
>
> A simple example illustrates this gap: In many real scenarios—such as multi-step planning, iterative web search, or coding tasks—an AI agent must (1) solve one subproblem, (2) use its solution as input for the next step, and (3) continue this process until the final goal is reached. Failing at any earlier step propagates errors downstream. Existing datasets cannot evaluate such behavior because they provide only one-off, independent questions.
>
> To bridge this, R-HORIZON creates **explicit cross-problem dependencies** that require models to solve multiple problems sequentially and maintain logical consistency across steps. We also include more realistic tasks such as **web search and agentic environments**, where long-horizon dependencies naturally occur. This keeps the benchmark **verifiable and controllable**, which is difficult to achieve with real-world long-horizon data due to the high cost and the lack of step-level ground truth.
>
> In summary, although no benchmark can fully replicate real-world complexity, R-HORIZON moves substantially closer by introducing structured multi-step dependencies and offering the first scalable and verifiable framework for evaluating long-horizon reasoning.
>
> ---
>
> ### **Response to Weakness on Imprecise Writing in the Evaluation Section**
>
> Thank you for your careful reading. We have fixed the above issues in the revised version (with the changes highlighted in blue). The details are as follows:
>
> - Section 4.1
>   - AMC23 is not discussed ->  $n \in \\\{1, 2, 4, 6, 8\\\}$ for AMC23.
>   - for MATH500 it seems that there is also data for 20 composed query numbers -> $n \in \\\{1, 2, 4, 8, 16, 20\\\}$ for MATH500.
> - 4.2, first paragraph: "R1-Qwen-7B drops from 93.6% (n = 1) to 0% (n = 16), which is 34.1% more than the 32B model" - reference is unclear (probably MATH500); Where does the 34.1% come from?
>   - "R1-Qwen-7B drops from 93.6% (n = 1) to 0% (n = 16), which is 34.1% more than the 32B model"-> For example, R1-Qwen-7B drops from 93.6% (n=1) to 0% (n=16), which is 34.1% more than the R1-Qwen-32B model.
>   - We apologize for not specifying the exact 32B model earlier. For clarity: R1-Qwen-7B drops from 93.6% (n = 1) to 0% (n = 16), while R1-Qwen-32B drops from 97.5% (n = 1) to 38% (n = 16). This means the 7B model experiences a 34.1% larger decline compared to the 32B model.
> - 5.1, Effective Reasoning Length of LRMs:
>   - "gap between the actual accuracy and theoretical accuracy of models becomes increasingly larger" - not necessarily true on AIME24->This exception occurs because the model’s performance on AIME24 is already very low when the number of queries reaches 3, leaving little room for the gap to increase further.
>   - "7B model’s error range is (4-6k tokens) while the 32B model’s error range is (8-10k tokens)" - context is missing, probably for MATH500-> "7B model’s error range is (4-6k tokens) while the 32B model’s error range is (8-10k tokens) for Math500"
> - Reflection is not defined or at least explained, which also makes it unclear why it matters.
>   - **Reflection** refers to the model’s self-reflective behaviors. For example, when the model produces phrases such as *“wait,” “let me…,” “but…”* and similar expressions, we define these as instances of **Reflection**. We have added this definition in the *Reflection Frequency and Depth of LRMs* section.
>   - The emergence of reflective behaviors indicates that the model is engaging in stronger exploratory reasoning, which can be considered a healthy thinking pattern. As the number of queries increases, the frequency of such reflections is expected to rise accordingly, and the span of each reflective segment should also become longer.
> - Sometimes "expected accuracy" and sometimes "theoretical accuracy" is used? I assume they are the same thing, so I suggest to use one term consistently.->We have unified the terminology and now consistently use **“Expected Accuracy”** throughout the paper.
> - Appendix D.3: "Figure 14 (b) and (c) show that all models fail to allocate thinking budget reasonably according to problem difficulty" - seems to be somewhat not true for R1-Qwen-7B -> Figure 14 (b) and (c) show that DeepSeek-R1, R1-Qwen-32B fail to allocate thinking budget reasonably according to problem difficulty"

---

> ### Author Response · Authors · 2025-11-19
> **Response to Reviewer hTcg (Part 2)**
>
> ### **Response to Weakness on Minor Issues**
>
> Thank you for pointing out our  typos and minor issues. We have fixed the above issues in the revision. The details are as follows:
>
> - introduction: no reference for CoT-> We have added relevant citations.
> - Figure 1: Theoretical Accuracy is not introduced at this point. (only on page 4)-> add reference to expected accuracy in introduction
> - 2.2:"Su et al. (2025); Yang et al. (2025b); Wu et al. (2025b) investigates..." - It should be "investigate", since it is a plural.-> We have fixed this.
> - Algorithm 1: "and Create" - lowercase create? -> We have fixed this.
> - 4.1: I believe, there is no whitespace before a footnote.-> There is no whitespace before the footnote.
> - 4.3, Impact of Number of Composed Queries and Different Reward Schemes: n=1 is not really a composed problem -> n=1 is our baseline, we mainly focus on composed problems.
> - Figure 5: not very readable in terms of fontsize as well as chosen color for example for early stop and output truncation - I had to zoom in to 400% to find an instance for output truncation -> We attempted to adjust the colors, but because the number of truncated outputs is very small, it is still difficult to visualize clearly.
>   - similar issue for Figure 6, especially the theoretical accuracy as well as Figures 7 to 9-> We have already increased the font size in the figure as much as possible.
> - Figure 6: missing whitespace before the brackets in the subfigure titles (also for Figure 9) -> We have added whitespace before the brackets in Figure 6 and Figure 9.
> - sometimes it is "Math500" and sometimes it is "MATH500" in the evaluation figures-> We change the "MATH500" to "Math500" in Figure 6 and now consistently use "Math500" throughout the paper.
> - Figure 8: x axis label might be misleading, maybe something like "2 Queries" or just "2" and so on would be better? ->  "Query 2" to "2" in Figure 8.
> - Appendix F.1: "policy loss 5 with" - should probably be something like "policy loss (see Eq. 5) with"-> "policy loss 5 with" to "policy loss (see Eq. 5) with" in Appendix F.1
> - Appendix H: details on model, which generated the responses are missing, where the questions are coming from, etc.-> We use models trained with n=1 and n=4 data in Section 4.3 and the example prompt is coming from AIME24, we have clarify this in Appendix H.
> - references:
>   - consider the proper capitalization of the titles, at least for proper names and abbreviations to improve readability -> We have corrected the improperly capitalized titles in Section 4.2.
>   - place of publication of arXiv references can only surmised from the URL -> We exported the BibTeX citations directly from arXiv, and that is why they appear in the current format. We will try to correct them in the revision.
>   - consider adding the access date for web references -> We added the corresponding access dates for the following web references: DeepScaleR, DeepCoder, Skywork-or1, and POLARIS.
>   - [Guo et al. 2025] - cited differently than the other arXiv references-> We have checked this citation and find it is the same as the BibTeX citation on arXiv.
>   - [Luo et al. 2025a/b] - URL is not a proper link-> The URL is strange, but it is the proper link to Notion blog.
>   - [MAA 2024/2025] - consider using something like the xurl package to fix the formatting-> [MAA 2024/2025] -> [AIME, 2024/2025]
>   - [Muennighoff et al. 2025] - cited differently than the other arXiv references - maybe this could be the blueprint for the others-> We have fixed this.
>
> ---
>
> ### **Response to Questions:**
>
> > Figure 2: What are the circles with a red outline?
>
> The red-outlined circles indicate dependencies, and Figure 2(b) provides an explanation of how dependencies are created.
>
> > 4.2, last paragraph: Why is AIME25 more challenging than AIME24?
>
> We made this statement mainly because, in Figure 3 (*Evaluation Results of R-HORIZON Benchmark*), most models perform much worse on AIME25 compared to AIME24.
>
> > Figure 6: How can the error position be above the token length? Because wrong answers tend to be longer for smaller query numbers?
>
> Yes. In Figure 6 (Query Num = 1), incorrect problems are generally longer than correct ones, but the number of incorrect answers on Math500 is very small compared to the number of correct answers (32 wrong vs 468 correct on R1-Qwen-7B). Since the average error position is higher than the average length of correct problems, this leads to the error position appearing above the average token length.
>
> > How much can the six benchmarks (MATH500, AMC23, AIME24/25, LiveCodeBench and WebShaper) be considered out of distribution compared to the training dataset based on Skywork-RL?
>
> For MATH500, AMC23, and AIME24/25, since they are all mathematical problems constructed in a similar way, we consider them in-distribution relative to the training data. In contrast, LiveCodeBench and WebShaper differ significantly in both format and construction, and are thus considered out-of-distribution.

---

> ### Author Response · Authors · 2025-11-27
> **Follow up Response to Reviewer hTcg**
>
> Dear Reviewer hTcg,
>
> We would like to express our sincere gratitude for your thorough review of our manuscript and for your valuable insights. We have carefully considered your feedback and have made significant revisions to address your concerns.
>
> In particular, we have:
>
> - Clarified the motivation of R-HORIZON by emphasizing realistic long-horizon reasoning scenarios and the need for interdependent problem-solving.
>
> - Corrected ambiguities and inconsistencies in the evaluation section, including explicit references, dataset details, definitions (e.g., Reflection), and unified terminology such as “Expected Accuracy.”
>
> - Fixed minor issues in figures, formatting, citations, and appendix details to improve readability and clarity.
>
> We kindly ask if you could take a moment to review our updated manuscript. If there is anything further that you would like us to clarify or any additional feedback you wish to provide, please let us know. We are more than willing to provide any additional information or make further revisions as needed.
>
> Thank you once again for your time and thoughtful feedback. Your insights have been instrumental in improving our paper.
>
> Sincerely,
>
> Authors of Submission 6849

---

### Author Response · Authors · 2025-12-03
**General response**

Dear Area Chair and All Reviewers,

We want to express our sincere gratitude for the constructive feedback provided by the reviewers for Submission 6849. We have carefully addressed each of the reviewer's concerns to enhance the quality and robustness of our paper. We also have revised our manuscript to address all reviewers' concerns. **All modifications in the revised manuscript are highlighted in blue**.

In this General response, we provide a summary of the key actions we have undertaken in response to the reviewers' comments. To help reduce your reviewing workload, we also provide a summary of the rebuttal process for our paper.

---

## Summary of the rebuttal process

In Summary, during the Author–Reviewer Discussion phase, **only 1/4 reviewers provided feedback, confirmed that we addressed their concerns and increase the score of our paper.**

1. **Reviewer hTcg: Rating: 6 / Confidence: 3**
   - **No follow-up**: Reviewer hTcg raised a weakness concerning that the dependencies are *artificial*, and also pointed out 8 inaccuracies in the paper as well as 13 minor issues. We responded to all suggestions one by one, and revised the corresponding sections of our paper accordingly.
2. **Reviewer JsjS: Rating: 6 / Confidence: 5**
   - **No follow-up**: Reviewer JsjS raised three highly insightful questions, including analyzing entropy decrease during training, evaluating rollout efficiency, and using compositional data for SFT. We conducted corresponding supplementary experiments and obtained additional insights.
3. **Reviewer sdPJ: Rating: 6 / Confidence: 3**
   - **No follow-up**: Reviewer sdPJ expressed concerns that the differences between the composition methods are relatively small and that the evaluation lacks closed-source models. We provided clarifications accordingly, and included results from Gemini-2.5 Pro.
4. **Reviewer 91MQ: Rating: 6 / Soundness: 3 → Rating: 8 / Soundness: 4**
   - **Reviewer follow-up (22 Nov)**: Reviewer 91MQ confirmed that most of the questions are were **resolved**. As a result, the reviewer is happy to increase the score and the soundness assessment of this paper.

---

## Clarifications

1. **The gap between R-HORIZON and realistic reasoning scenarios** (Reviewer hTcg & JsjS)
   - We clarified the motivation behind R-HORIZON, emphasizing its aim to bridge the gap between single-step benchmarks and realistic long-horizon reasoning scenarios.
2. **Distinction from GSM-Infinite**  (Reviewer JsjS)
   - We distinguished R-HORIZON from GSM-Infinite in both methodology (cross-problem composition vs. intra-problem substep composition) and evaluation focus (long output trajectories vs. long input contexts).
3. **The impact of different composition method** (Reviewer sdPJ)
   - We clarified the differences between directly composition and sequentially composition strategy. Even minimal sequential dependencies substantially increase model error rates compared to independent compositions or single-step queries.
4. **Different composition method on math, code and agentic task** (Reviewer 91MQ)
   - R-HORIZON aims to simulate *realistic* long-horizon reasoning rather than force uniform dependencies across all tasks, we design composition strategies according to the characteristics of each domain.
5. **Reward Design for all and last problem** (Reviewer 91MQ)
   - We demonstrate  the model can sometimes answer later questions correctly even after failing earlier ones and we report the discrepancy between Acc_last and Acc_all in ablation study.

## Supplementary Experiments

1. **Analysis of entropy decrease as the number of composed problem increases**  (Reviewer JsjS)
   - We provided a detailed analysis of entropy reduction during RL training with increasing number of problem composition, including reflection token statistics.
2. **Impact of Supervised Fine-tuning (SFT) on R-HORIZON Performance** (Reviewer JsjS)
   - We included supervised fine-tuning results showing that exposure to compositional data substantially improves model performance on multi-step tasks.
3. **Analysis of rollout efficiency during RL process** (Reviewer JsjS)
   - We quantified how balanced reward signals in constructed datasets enhance rollout efficiency during RL and observe that using composed training data improves  rollout efficiency.
4. **Additional evaluation on closed-source models** (Reviewer sdPJ)
   - In addition to the original three closed-source model (Gemini-2.5 Flash, o4-mini, Claude-Sonnet-4) evaluations, we further included results from Gemini-2.5 Pro.
5. **Error Type Analysis Before & After the Training** (Reviewer 91MQ)
---

By implementing these revisions, we believe the enhancements address the reviewers' initial concerns and provide valuable insights to the field.

Thank you for taking the time to read our rebuttal. We are confident that the improvements made have significantly strengthened the quality and impact of our work.

Best regards,

Authors of submission 6849

---

### Meta-Review · Area_Chair_888o · 2025-12-29

**Summary:**

The paper proposes R-HORIZON, a benchmark and data construction framework for evaluating and improving long-horizon reasoning in large reasoning models (LRMs).  The authors addressed most reviews, added experiments, and revised the manuscript. One reviewer raised scores explicitly, and the work meets the acceptance criteria.

**Reviewer Concerns:**

Resolved concerns: R-HORIZON’s rationality, prior work distinction, writing ambiguities, closed-source gaps, etc.

Unresolved: some minor issues, not affecting core contributions.

**Reviewer Scores:**

Reviewers hTcg, JsjS, and sdPJ gave an initial 6 (above threshold), all concerns addressed. Reviewer 91MQ raised score from 6 to 8, soundness from 3 to 4. One explicit boost, all core issues solved, satisfying acceptance standards.

---

### Decision · Program_Chairs · 2026-01-26

Accept (Poster)